# ADVERSARIAL EXAMPLES ARE A NATURAL CONSEQUENCE OF TEST ERROR IN NOISE

## ABSTRACT

Over the last few years, the phenomenon of *adversarial examples* — maliciously constructed inputs that fool trained machine learning models — has captured the attention of the research community, especially when the adversary is restricted to making small modifications of a correctly handled input. At the same time, less surprisingly, image classifiers lack human-level performance on randomly corrupted images, such as images with additive Gaussian noise. In this work, we show that these are two manifestations of the same underlying phenomenon. We establish this connection in several ways. First, we find that adversarial examples exist at the same distance scales we would expect from a linear model with the same performance on corrupted images. Next, we show that Gaussian data augmentation during training improves robustness to small adversarial perturbations and that adversarial training improves robustness to several types of image corruptions. Finally, we present a model-independent upper bound on the distance from a *corrupted* image to its nearest error given test performance and show that in practice we already come close to achieving the bound, so that improving robustness further for the corrupted image distribution requires significantly reducing test error. All of this suggests that improving adversarial robustness should go hand in hand with improving performance in the presence of more general and realistic image corruptions. This yields a computationally tractable evaluation metric for defenses to consider: test error in noisy image distributions.

## 1 INTRODUCTION

State-of-the-art computer vision models can achieve superhuman performance on many image classification tasks. Despite this, these same models still lack the robustness of the human visual system to various forms of image corruptions. For example, they are distinctly subhuman when classifying images distorted with additive Gaussian noise (Dodge & Karam, 2017b), they lack robustness to different types of blur, pixelation, and changes in brightness (Hendrycks & Dietterich, 2018), lack robustness to random translations of the input (Azulay & Weiss, 2018), and even make errors when foreign objects are inserted into the field of view (Rosenfeld et al., 2018). At the same time, they also are sensitive to small, worst-case perturbations of the input, so-called "adversarial examples" (Szegedy et al., 2014). This latter phenomenon has struck many in the machine learning community as surprising and has attracted a great deal of research interest, while the former seems to inspire less surprise and has received considerably less attention.

Our classification models make errors on two different sorts of inputs: those found by randomly sampling from some predetermined distribution, and those found by an adversary deliberately searching for the closest error to a given point. In this work, we ask what, if anything, is the difference between these two types of error. Given that our classifiers make errors in these corrupted image distributions, there must be a closest such error; do we find that this closest error appears at the distance we would expect from the model's performance in noise, or is it in fact "surprisingly" close?

The answer to this question has strong implications for the way we approach the task of eliminating these two types of errors. An assumption underlying most of the work on adversarial examples is that solving it requires a different set of methods than the ones being developed to improve model generalization. The adversarial defense literature focuses primarily on improving robustness to small perturbations of the input and rarely reports improved generalization in *any* distribution.

We claim that, on the contrary, adversarial examples are found at the same distance scales that one should expect given the performance on noise that we see in practice. We explore the connection between small perturbation adversarial examples and test error in noise in two different ways.

First, in Sections 4 and 5, we provide empirical evidence of a close relationship between test performance in Gaussian noise and adversarial perturbations. We show that the errors we find close to the clean image and the errors we sample under Gaussian noise are part of the same large set and show some visualizations that illustrate this relationship. (This analysis builds upon prior work (Fawzi et al., 2018; 2016) which makes smoothness assumptions on the decision boundary to relate these two quantities.) This suggests that training procedures designed to improve adversarial robustness might reduce test error in noise and vice versa. We provide results from experiments which show that this is indeed the case: for every model we examined, either both quantities improved or neither did. In particular, a model trained on Gaussian noise shows significant improvements in adversarial robustness, comparable to (but not quite as strong as) a model trained on adversarial examples. We also found that an adversarially trained model on CIFAR-10 shows improved robustness to random image corruptions.

Finally, in Section 6, we establish a relationship between the error rate of an image classification model in the presence of Gaussian noise and the existence of adversarial examples for *noisy* versions of test set images. In this setting we can actually prove a rigorous, model-independent bound relating these two quantities that is achieved when the error set is a half space, and we see that the models we tested are already quite close to this optimum. Therefore, for these noisy image distributions, our models are already almost as adversarially robust as they can be given the error rates we see, so the only way to defend against adversarial examples is to reduce test error.

In this work we will investigate several different models trained on the MNIST, CIFAR-10 and ImageNet datasets. For MNIST and CIFAR-10 we look at the naturally trained and adversarially trained models which have been open-sourced by Madry et al. (2017). We also trained the same model on CIFAR-10 with Gaussian data augmentation. For ImageNet, we investigate Wide ResNet-50 trai]ned with Gaussian data augmentation. We were unable to study the effects of adversarial training on ImageNet because no robust open sourced model exists (we considered the models released in Tramèr et al. (2017) but found that they only minimally improve robustness to the white box PGD adversaries we consider here). Additional training details can be found in Appendix A.

## 2 RELATED WORK

The broader field of *adversarial machine learning* studies general ways in which an adversary may interact with an ML system, and dates back to 2004 (Dalvi et al., 2004; Biggio & Roli, 2018). Since the work of Szegedy et al. (2014), a subfield has focused specifically on the phenomenon of small adversarial perturbations of the input, or "adversarial examples." In Szegedy et al. (2014) it was proposed these adversarial examples occupy a dense, measure-zero subset of image space. However, more recent work has provided evidence that this is not true. For example, Fawzi et al. (2016); Franceschi et al. (2018) shows that under linearity assumptions of the decision boundary small adversarial perturbations exist when test error in noise is non-zero. Gilmer et al. (2018b) showed for a specific data distribution that there is a fundamental upper bound on adversarial robustness in terms of test error. Mahloujifar et al. (2018) has generalized these results to a much broader class of distributions.

Recent work has proven for a synthetic data distribution that adversarially robust generalization requires more data (Schmidt et al., 2018). The distribution they consider when proving this result is a mixture of high dimensional Gaussians. As we will soon discuss, every set $E$ of small measure in the high dimensional Gaussian distribution has large boundary measure. Therefore, at least for the data distribution considered, the main conclusion of this work, "adversarially robust generalization requires more data", is a direct corollary of the statement "generalization requires more data."

## 3 TEST ERROR AND ADVERSARIAL ROBUSTNESS

Understanding the relationship between nearby errors and model generalization requires understanding the geometry of the *error set* of a statistical classifier, that is, the set of points in the input space

on which the classifier makes an incorrect prediction. In particular, the assertion that these adversarial examples are a distinct phenomenon from test error is equivalent to stating that the error set is in some sense poorly behaved. We study two functions of a model's error set $E$.

The first quantity, **test error** under a given distribution of inputs $q(x)$, is the probability that a random sample from the distribution $q$ is in $E$. We will denote this $\mathbb{P}_{x \sim q}[x \in E]$; reducing this quantity when $q$ is the natural data distribution is the goal of supervised learning. While one usually takes $q$ to be the distribution from which the training set was sampled, we will also consider other distributions over the course of this paper.

When $q$ includes points from outside the natural data distribution, a decision needs to be made about the labels in order to define $E$. The only such cases we will consider in this paper are noisy perturbations of training or test points, and we will always assume that the noise is at a scale which is small enough not to change the label. This assumption is commonly made in works which study model robustness to random corruptions of the input (Hendrycks & Dietterich, 2018; Dodge & Karam, 2017b). Some examples noisy images can be found in Figure 7 in the appendix.

The second quantity is called **adversarial robustness**. For an input $x$ and a metric on the input space $d$, let $d(x, E)$ denote the distance from $x$ to the nearest point of $E$. For any $\epsilon$, let $E_\epsilon$ denote the set $\{x : d(x, E) < \epsilon\}$, the set of points within $\epsilon$ of an error. The adversarial robustness of the model is then $\mathbb{P}_{x \sim q}[x \in E_\epsilon]$, the probability that a random sample from $q$ is within distance $\epsilon$ of some point in the error set. Reducing this quantity is the goal of much of the adversarial defense literature. When we refer to "adversarial examples" in this paper, we will always mean these nearby errors.

In geometric terms we can think of $\mathbb{P}_{x \sim q}[x \in E]$ as a sort of volume of the error set while $\mathbb{P}_{x \sim q}[x \in E_\epsilon]$ is related to its surface area. More directly, $\mathbb{P}_{x \sim q}[x \in E_\epsilon]$ is what we will call the $\epsilon$-*boundary measure*, the volume under $q$ of the region within $\epsilon$ of the surface or the interior.

The adversarial example phenomenon is then simply that, for small $\epsilon$, $\mathbb{P}_{x \sim q}[x \in E_\epsilon]$ can be large even when $\mathbb{P}_{x \sim q}[x \in E]$ is small. In other words, most correctly classified inputs are very close to a misclassified point, even though the model is very accurate. In high-dimensional spaces this phenomenon is not isolated to the error sets of statistical classifiers. In fact *almost every* nonempty set of small volume has large $\epsilon$-boundary measure, even sets that seem very well-behaved. As a simple example, consider the measure of the set $E = \{x \in \mathbb{R}^n : ||x||_2 < 1\}$ under the Gaussian distribution $q = \mathcal{N}(0, \sigma^2 I)$. For $n = 1000$, $\sigma = 1.05/\sqrt{n}$, and $\epsilon = 0.1$, we have $\mathbb{P}_{x \sim q}[x \in E] \approx 0.02$ and $\mathbb{P}_{x \sim q}[x \in E_\epsilon] \approx 0.98$, so most samples from $q$ will be close to $E$ despite the fact that $E$ has relatively little measure under the Gaussian distribution. If we relied only on our low-dimensional spatial intuition, we might be surprised to find how consistently small adversarial perturbations could be found — 98% of our test points would have an error at distance 0.1 or less even though only 2% are misclassified.

In high dimensions, it is much easier for most points to be close to some set even if that set itself has a small volume. Contrary to what one might expect from our low-dimensional intuition, this does not require the set in question to be somehow pathological; in our example, it was just a ball. Therefore, when we see that some image classifier has errors in some noise distribution $q$ (so that $\mathbb{P}_{x \sim q}[x \in E]$ is appreciably bigger than zero) it is possible that $E_\epsilon$ is much larger even if $E$ is quite simple, so the existence of small worst-case perturbations should be expected given imperfect robustness to large average-case corruptions. In the sections that follow we will make this precise.

## 4 Errors in Noise Suggest Adversarial Examples for Clean Images

**The Linear Case.** For linear models, the relationship between errors in Gaussian noise and small perturbations of a clean image is exact. For an image $x$, let $d(x)$ be the distance from $x$ to decision boundary and let $\sigma(x, \mu)$ be the $\sigma$ for which $\mathbb{P}_{x \sim q}[x \in E]$ is some fixed error rate $\mu$. (As we mentioned in the introduction, we assume that $\sigma$ is small enough that adding this noise does not change the "correct" label.) Then we have $d(x) = -\sigma(x, \mu)\Phi^{-1}(\mu)$, where

$$\Phi(t) = \frac{1}{\sqrt{2\pi}} \int_{-\infty}^{t} \exp(-x^2/2)dx$$

is the cdf of the univariate standard normal distribution.

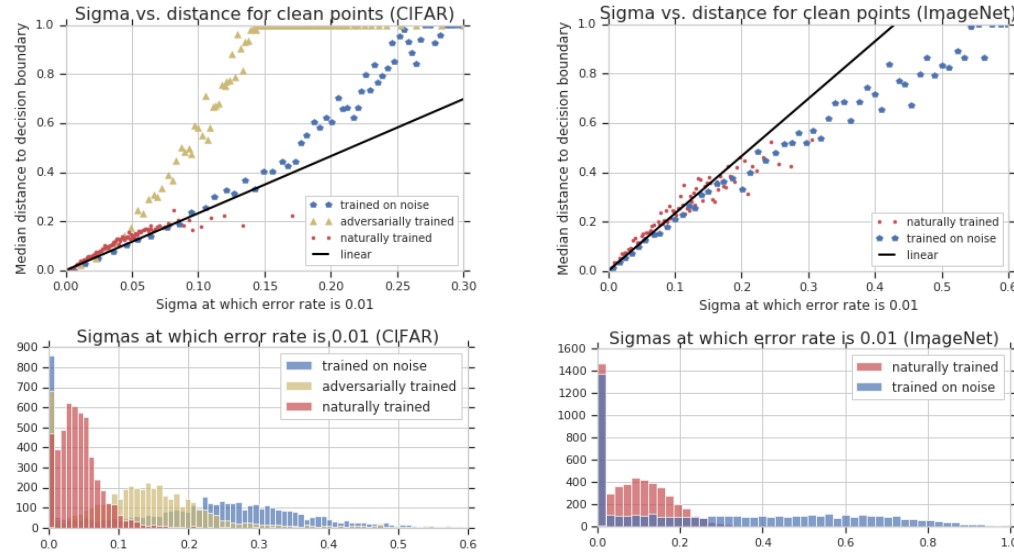

Figure 1: Comparing the distance to decision boundary with the $\sigma$ for which the error rate in Gaussian noise is 1%. Each point represents 50 images from the test set, and the median values for each coordinate are shown. (The PGD attack was run with $\epsilon = 1$, so the distances to the decision boundary reported here are cut off at 1.) We also see histograms of the $x$ coordinates. (A misclassified point is assigned $\sigma = 0$.)

Note that this equality depends only on the error rate $\mu$ and the standard deviation $\sigma$ of a single component, and not directly on the dimension. This might seem at odds with the emphasis on high-dimensional geometry in Section 3. The dimension does appear if we consider the norm of a typical sample from $\mathcal{N}(0, \sigma^2 I)$, which is $\sigma\sqrt{n}$. As the dimension increases, so does the ratio between the distance to a noisy image and the distance to the decision boundary.

The decision boundary of a neural network is, of course, not linear. However, by computing the ratio between $d(x)$ and $\sigma(x, \mu)$ for neural networks and comparing it to what it would be for a linear model, we can investigate the question posed in the introduction: do we see adversarial examples at the distances we do because of pathologies in the shape of the error set, or do we find them at about the distances we would expect given the error rates we see in noise? We ran experiments on the error sets of several neural image classifiers and found evidence that is much more consistent with the second of these two possibilities. This relationship was also explored in Fawzi et al. (2016; 2018); here we additionally measure how data augmentation affects this relationship.

We examined this relationship for neural networks when $\mu = 0.01$. For each test point, we compared $\sigma(x, \mu)$ to an estimate of $d(x)$. It is not actually possible to compute $d(x)$ precisely for the error set of a neural network. In fact, finding the distance to the nearest error is NP-hard (Katz et al., 2017). Instead, the best we can do is to search for an error using a method like PGD (Madry et al., 2017) and report the nearest error we can find.

Figure 1 shows the results for several CIFAR-10 and ImageNet models, including ordinary trained models, models trained on noise with $\sigma = 0.4$, and an adversarially trained CIFAR-10 model. We also included a line representing how these quantities would be related for a linear model.

We can see that none of the models we examined have nearby errors at a scale much smaller than we would expect from a linear model. Indeed, while the adversarially trained model does deviate from the linear case to a greater extent than the others, it does so in the direction of *greater* distances to the decision boundary. Moreover, we can see from the histograms that both of the interventions that increase $d(x)$ also increase $\sigma(x, \mu)$. So, to explain the distances to the errors we can find using PGD, it is not necessary to rely on any great complexity in the shape of the error set; a linear model with the same error rates in noise would have errors just as close.

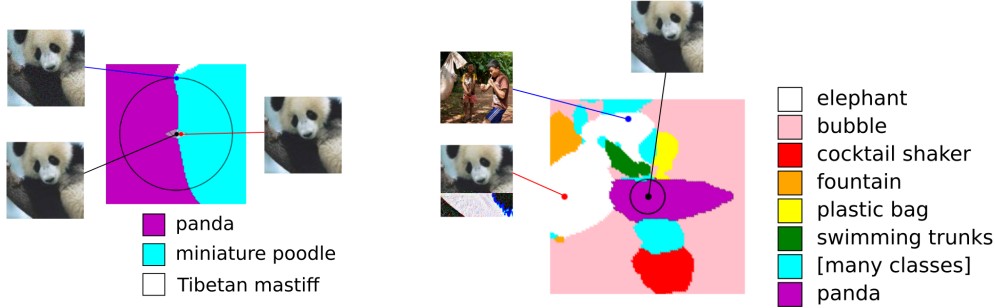

Figure 2: Two-dimensional slices of image space through different triples of points together with the classes assigned by a trained model. The black circle in both images has radius $31.4$, corresponding to noise with $\sigma = 31.4/\sqrt{n} = 0.08$.
*Left:* An image from the test set (black), a random misclassified Gaussian perturbation at standard deviation $0.08$ (blue), and an error found using PGD (red). The estimated measure of the cyan region ("miniature poodle") in the Gaussian distribution is about $0.1\%$. The small diamond-shaped region in the center of the image is the $l_\infty$ ball of radius 8/255.
*Right:* A slice at a larger scale with the same black point, together with an error from the clean set (blue) and an adversarially constructed error (red) which are both assigned to the same class ("elephant").

**Visualizing the Decision Boundary.** In Figure 2 we drew some pictures of two-dimensional slices of image space through several different triples of points. (Similar visualizations have previously appeared in Fawzi et al. (2018), and are called "church window plots.")

We see some common themes. In the figure on the left, we see that an error found in Gaussian noise lies in the same connected component of the error set as an error found using PGD, and that at this scale that component visually resembles a half space. This figure also illustrates the relationship between test error and adversarial robustness. To measure adversarial robustness is to ask whether or not there are any errors in the $l_\infty$ ball — the small diamond-shaped region in the center of the image — and to measure test error in noise is to measure the volume of the error set in the defined noise distribution. At least in this slice, nothing distinguishes the PGD error from any other point in the error set apart from its proximity to the center point.

The figure on the right shows a different slice through the same test point but at a larger scale. This slice includes an ordinary test error along with an adversarial perturbation of the center image constructed with the goal of maintaining visual similarity while having a large $l_2$ distance. The two errors are both classified (incorrectly) by the model as "elephant." This adversarial error is actually *farther* from the center than the test error, but they still clearly belong to the same connected component. This suggests that defending against worst-case content-preserving perturbations (Gilmer et al., 2018a) requires removing all errors at a scale comparable to the distance between unrelated pairs of images. Many more church window plots can be found in Appendix G.

## 5 COMPARING ADVERSARIAL TRAINING TO TRAINING ON NOISE

For a linear model, improving generalization in the presence of noise is equivalent to increasing the distance to the decision boundary. The results from the previous section suggest that a similar relationship should hold for other statistical classifiers, including neural networks. That is, augmenting the training data distribution with noisy images ought to increase the distance to the decision boundary, and augmenting the training distribution with small-perturbation adversarial examples should improve performance in noise. Here we present evidence that this is the case.

We analyzed the performance of the models described in Section 1 on four different noise distributions: two types of Gaussian noise, pepper noise (Hendrycks & Dietterich, 2018), and a randomized variant of the stAdv adversarial attack introduced in Xiao et al. (2018). We used both ordinary, spherical Gaussian noise and what we call "PCA noise," which is Gaussian noise supported only on the

| Dataset | CIFAR-10 | | | | ImageNet | | |
|---|---|---|---|---|---|---|---|
| Training | Vanilla | Noise $\sigma = 0.1$ | Noise $\sigma = 0.4$ | Adv | Vanilla | Noise $\sigma = 0.4$ | Noise $\sigma = 0.8$ |
| **Noise Type** | | | | | | | |
| Clean | 95.0% | 93.5% | 84.0% | 87.3% | 76.0% | 74.4% | 72.6% |
| PCA100, $\sigma = 0.2$ | 93.2% | 92.3% | 83.6% | 86.5% | 45.5% | 56.5% | 59.7% |
| PCA100, $\sigma = 0.4$ | 82.6% | 83.1% | 81.0% | 80.6% | 13.5% | 17.7% | 19.7% |
| Pepper, $p = 0.1$ | 20.2% | 53.3% | 81.2% | 38.4% | 31.3% | 70.0% | 69.1% |
| Pepper, $p = 0.3$ | 12.3% | 18.9% | 58.0% | 21.1% | 5.4% | 56.0% | 61.5% |
| Gaussian, $\sigma = 0.1$ | 29.1% | 89.0% | 85.1% | 77.8% | 60.7% | 73.3% | 71.7% |
| Gaussian, $\sigma = 0.2$ | 13.5% | 38.8% | 83.5% | 42.1% | 27.9% | 70.5% | 69.3% |
| stAdv, $\sigma = 0.5$ | 52.3% | 84.4% | 77.9% | 81.7% | 57.3% | 67.3% | 69.0% |
| stAdv, $\sigma = 2.0$ | 17.4% | 30.6% | 52.1% | 27.0% | 11.4% | 27.2% | 31.3% |
| $l_p$ **robustness** | | | | | | | |
| $l_2, \epsilon = 0.5$ | 0.3% | 39.2% | 54.5% | 58.3% | 7.9% | 43.8% | 47.7% |
| $l_2, \epsilon = 1.0$ | 0.0% | 9.5% | 25.1% | 29.7% | 0.5% | 16.8% | 22.5% |
| $l_\infty, \epsilon = 1/255$ | 26.2% | 84.4% | 76.6% | 83.5% | 0.8% | 20.1% | 25.0% |
| $l_\infty, \epsilon = 4/255$ | 0.4% | 39.8% | 49.6% | 68.3% | 0.0% | 0.1% | 0.1% |
| $l_\infty, \epsilon = 8/255$ | 0.0% | 10.3% | 20.0% | 45.4% | 0.0% | 0.0% | 0.0% |

Table 1: The performance of the models we considered under various noise distributions, together with our measurements of those models' robustness to small $l_p$ perturbations. For all the robustness tests we used PGD with 100 steps and a step size of $\epsilon/25$. The adversarially trained CIFAR-10 model is the open sourced model from Madry et al. (2017).

subspace spanned by the first 100 principal components of the training set. Pepper noise randomly assigns channels of the image to 1 with some fixed probability. Details of the stAdv attack can be found in Appendix B, but it visually similar to Gaussian blurring where $\sigma$ controls the severity of the blurring. Example images that have undergone each of the noise transformations we used can be found in Appendix I. Each model was also tested for $l_p$ robustness with a variety of norms and $\epsilon$'s using the same PGD attack as in Section 4.

For CIFAR-10, standard Gaussian data augmentation yields comparable (but slightly worse) results to adversarial training on all considered metrics. For ImageNet we found that Gaussian data augmentation improves robustness to small $l_2$ perturbations as well as robustness to other noise corruptions. The results are shown in Table 1. This holds both for generalization in all noises considered and for robustness to small perturbations. We found that performing data augmentation with heavy Gaussian noise ($\sigma = 0.4$ for CIFAR-10 and $\sigma = 0.8$ for ImageNet) worked best. The adversarially trained CIFAR-10 models were trained in the $l_\infty$ metric and they performed especially well on worst-case perturbations in this metric. Prior work has observed that Gaussian data augmentation helps small perturbation robustness on MNIST (Kannan et al., 2018), but to our knowledge we are the first to measure this on CIFAR-10 and ImageNet.

Neither augmentation method shows much improved generalization in PCA noise. We hypothesize that adversarially trained models learn to project away the high-frequency information in the input, which would do little to improve performance in PCA noise, which is supported in the low-frequency subspace of the data distribution. Further work would be required to establish this.

We also considered the MNIST adversarially trained model from Madry et al. (2017), and found it to be a special case where although robustness to small perturbations was increased generalization in noise was not improved. This is because this model violates the linearity assumption discussed in Section 4. This overfitting to the $l_\infty$ metric has been observed in prior work (Sharma & Chen, 2017). More details can be found in Appendix D.

Although no $l_p$-robust open sourced ImageNet model exists, recent work has found that the adversarially trained models on Tiny ImageNet from Kannan et al. (2018) generalize very well on a large suite of common image corruptions (Hendrycks & Dietterich, 2018).

**Failed Adversarial Defenses Do Not Improve Generalization in Noise.** We performed a similar analysis on seven previously published adversarial defense strategies. These methods have already been shown to result in masking gradients, which cause standard optimization procedures to fail to find errors, rather than actually improving small perturbation robustness (Athalye et al., 2018). We find

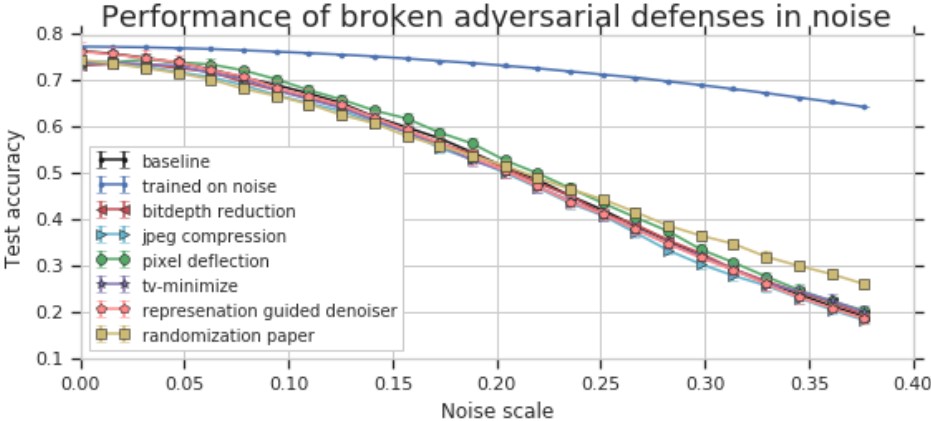

Figure 3: The performance in Gaussian noise of several previously published defenses for ImageNet, along with a model trained on Gaussian noise at $\sigma = 0.4$ for comparison. For each point we ran ten trials; the error bars show one standard deviation. All of these defenses are now known not to improve adversarial robustness (Athalye et al., 2018). The defense strategies include bitdepth reduction (Guo et al., 2017), JPEG compression (Guo et al., 2017; Dziugaite et al., 2016; Liu et al., 2018; Aydemir et al., 2018; Das et al., 2018; 2017), Pixel Deflection (Prakash et al., 2018), total variance minimization (Guo et al., 2017), respresentation-guided denoising (Liao et al., 2018), and random resizing and random padding of the input image (Xie et al., 2017).

that these methods also show no improved generalization in Gaussian noise. The results are shown in Figure 3. Given how easy it is for a method to show improved robustness to standard optimization procedures without changing the decision boundary in any meaningful way, we strongly recommend that future defense efforts evaluate on out-of-distribution inputs such as the noise distributions we consider here. The current standard practice of evaluating solely on gradient-based attack algorithms is making progress more difficult to measure.

**Obtaining Zero Test Error in Noise is Nontrivial.** It is important to note that applying Gaussian data augmentation does not reduce error rates in Gaussian noise to zero. For example, we performed Gaussian data augmentation on CIFAR-10 at $\sigma = .15$ and obtained 99.9% training accuracy but 77.5% test accuracy in the same noise distribution. (For comparison, the naturally trained obtains 95% clean test accuracy.) Previous work (Dodge & Karam, 2017b) has also observed that obtaining perfect generalization in large Gaussian noise is nontrivial. This mirrors Schmidt et al. (2018), which found that small perturbation robustness did not generalize to the test set. This is perhaps not surprising given that error rates on the *clean* test set are also non-zero. Although the model is in some sense "superhuman" with respect to clean test accuracy, it still makes many mistakes on the clean test set that a human would never make. We collected some examples in Appendix I. More detailed results on training and testing in noise can be found in Appendices C and H.

## 6 ERRORS IN NOISE IMPLY ADVERSARIAL EXAMPLES FOR NOISY IMAGES

**The Gaussian Isoperimetric Inequality.** Let $x$ be a correctly classified image and consider the distribution $q$ of Gaussian perturbations of $x$ with some fixed variance $\sigma^2 I$. For this distribution, there is a precise sense in which small adversarial perturbations exist only because test error is nonzero. That is, given the error rates we actually observe on noisy images, most noisy images *must* be close to the error set. This result holds completely independently of any assumptions about the model and follows from a fundamental geometric property of the high-dimensional Gaussian distribution, which we will now make precise.

For an image $x$ and the corresponding noisy image distribution $q$, let $\epsilon_q^*(E)$ be the median distance from one of these noisy images to the nearest error. (In other words, it is the $\epsilon$ for which $\mathbb{P}_{x \sim q}[x \in E_\epsilon] = \frac{1}{2}$.) As before, let $\mathbb{P}_{x \sim q}[x \in E]$ be the probability that a random Gaussian perturbation

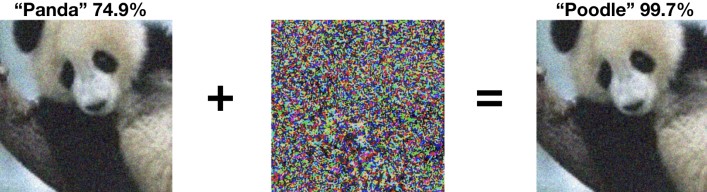

Figure 4: The adversarial example phenomenon occurs for noisy images as well as clean ones. Starting with a noisy image that that is correctly classified, one can apply carefully crafted imperceptible noise to it which causes the model to output an incorrect answer. This occurs even though the error rate among random Gaussian perturbations of this image is small (less than .1% for the ImageNet panda shown above). In fact, we prove that the presence of errors in Gaussian noise *logically implies* that small adversarial perturbations exists around noisy images. The only way to "defend" against such adversarial perturbations is to reduce the error rate in Gaussian noise.

of $x$ lies in $E$. It is possible to deduce a bound relating these two quantities from the *Gaussian isoperimetric inequality* (Borell, 1975). The form we will use is:

**Theorem** (Gaussian Isoperimetric Inequality). *Let $q = \mathcal{N}(0, \sigma^2 I)$ be the Gaussian distribution on $\mathbb{R}^n$ with variance $\sigma^2 I$, and let $\mu = \mathbb{P}_{x \sim q}[x \in E]$.*

*Write $\Phi(t) = \frac{1}{\sqrt{2\pi}} \int_{-\infty}^{t} \exp(-x^2/2)dx$, the cdf of the univariate standard normal distribution. If $\mu \geq \frac{1}{2}$, then $\epsilon_q^*(E) = 0$. Otherwise, $\epsilon_q^*(E) \leq -\sigma \Phi^{-1}(\mu)$, with equality when $E$ is a half space.*

In particular, for any machine learning model for which the error rate in the distribution $q$ is at least $\mu$, the median distance to the nearest error is at most $-\sigma \Phi^{-1}(\mu)$. (Note that $\Phi^{-1}(\mu)$ is negative when $\mu < \frac{1}{2}$.) Because each coordinate of a multivariate normal is a univariate normal, $-\Phi^{-1}(\mu)$ is the distance to a half space for which the error rate is $\mu$ when $\sigma = 1$. (We have the same indirect dependence on dimension here as we saw in Section 4: the distance to a typical sample from the Gaussian is $\sigma\sqrt{n}$.)

In Appendix E we will give the more common statement of the Gaussian isoperimetric inequality along with a proof of the version presented here. In geometric terms, we can say that a half space is the set $E$ of a fixed volume that minimizes the surface area under the Gaussian measure, similar to how a circle is the set of fixed area that minimizes the perimeter. So among models with some fixed test error $\mathbb{P}_{x \sim q}[x \in E]$, the most robust on this distribution are the ones whose error set is a half space.

**Comparing Neural Networks to the Isoperimetric Bound.** We evaluated these quantities for several models and many images from the CIFAR-10 and ImageNet test sets. Just like for clean images, we found that most noisy images are both correctly classified and very close to a visually similar image which is not. (See Figure 4.)

As we mentioned in Section 4, it is not actually possible to compute $\epsilon_q^*$ precisely for the error set of a neural network, so we again report an estimate. For each test image, we took 1,000 samples from the corresponding Gaussian and estimated $\epsilon_q^*$ using PGD with 200 steps on each sample and reported the median.

We find that for the five models we considered on CIFAR-10 and ImageNet, the relationship between our estimate of $\epsilon_q^*(E)$ and $\mathbb{P}_{x \sim q}[x \in E]$ is already close to optimal. This is visualized in Figure 5. Note that in both cases, adversarial training does improve robustness to small perturbations, but the gains are primarily because error rates in Gaussian noise were dramatically improved, and less because the surface area of the error set was decreased. In particular, many test points do not appear on these graphs because error rates in noise were so low that we did not find any errors among the 100,000 samples we used. For example, for the naturally trained CIFAR model, about 1% of the points lie off the left edge of the plot, compared to about 59% for the adversarially trained model and 70% for the model trained on noise. This shows that adversarial training on small perturbations improved generalization to large random perturbations, as the isoperimetric inequality says it must.

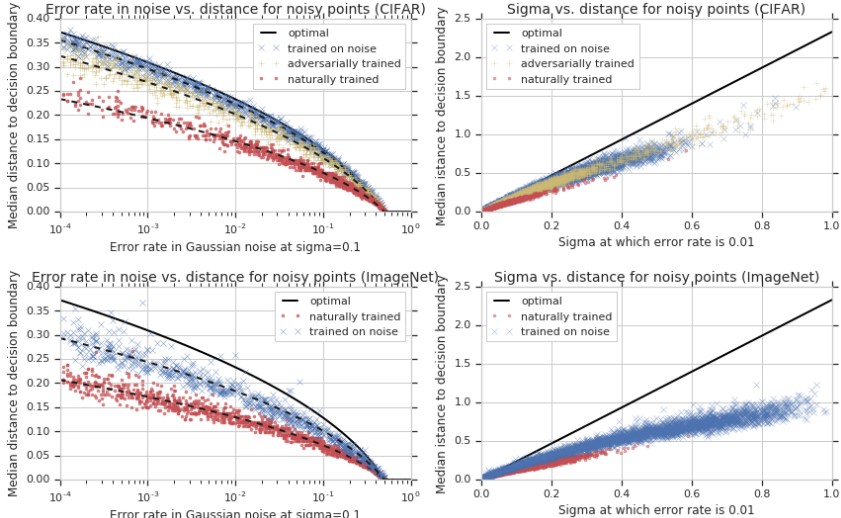

Figure 5: These plots give two ways to visualize the relationship between the error rate in noise and the distance from noisy points to the decision boundary (found using PGD). Each point on each plot represents one image from the test set. On the left, we compare the error rate of the model on Gaussian perturbations at $\sigma = 0.1$ to the distance from the median *noisy* point to its nearest error. On the right, we compare the $\sigma$ at which the error rate is $0.01$ to this same median distance. (The plots on the right are therefore similar to the plots in Figure 1.) The thick black line at the top of each plot is the upper bound provided by the Gaussian isoperimetric inequality. We include data from a model trained on clean images, an adversarially trained model, and a model trained on Gaussian noise ($\sigma = 0.4$.) As mentioned in Section 1, we were unable to run this experiment on an adversarially robust ImageNet model.

Not all models or functions will be this close to optimal. As a simple example, if we took one of the CIFAR models shown in Figure 5 and modified it so that the model outputs an error whenever each coordinate of the input is an integer multiple of $10^{-6}$, the resulting model would have an error within $\sqrt{\frac{1}{2} \cdot 10^{-6} \cdot \dim(\text{CIFAR})} \approx 0.039$ of every point. In this case, adversarial examples *would* be a distinct phenomenon from test performance, since $\epsilon_q^*(E)$ would be far from optimal.

The contrast between these two settings is important for adversarial defense design. If adversarial examples arose from a badly behaved decision boundary (as in the latter case), then it would make sense to design defenses which attempt to smooth out the decision boundary in some way. However, because we observe that image models are already close to the optimal bound on robustness for a fixed error rate in noise, future defense design should attempt to improve generalization in noise. Currently there is a considerable subset of the adversarial defense literature which develops methods that would remove any small "pockets" of errors but which don't improve model generalization. One example is Xie et al. (2017) which proposes randomly resizing the input to the network as a defense strategy. Unfortunately, this defense, like many others, has been shown to be ineffective against stronger adversaries (Carlini & Wagner, 2017a;b; Athalye et al., 2018).

## 7 CONCLUSION

We proved a fundamental relationship between generalization in noisy image distributions and the existence of small adversarial perturbations. By appealing to the Gaussian isoperimetric inequality, we formalized the notion of what it means for a decision boundary to be badly behaved. We showed that, for noisy images, there is very little room to improve robustness without also decreasing the volume of the error set, and we provided evidence that small perturbations of clean images can also be explained in a similar way. These results show that small-perturbation adversarial robustness is closely related to generalization in the presence of noise and that future defense efforts can measure progress by measuring test error in different noise distributions.

Indeed, several such noise distributions have already been proposed, and other researchers have developed methods which improve generalization in these distributions (Hendrycks & Dieterich, 2018; Dodge & Karam, 2017b;a; Vasiljevic et al., 2016; Zheng et al., 2016). Our work suggests that adversarial defense and improving generalization in noise involve attacking the same set of errors in two different ways — the first community tries to remove the errors on the boundary of the error set while the second community tries to reduce the volume of the error set. The isoperimetric inequality connects these two perspectives, and suggests that improvements in adversarial robustness should result in improved generalization in noise and vice versa. Adversarial training on small perturbations on CIFAR-10 also improved generalization in noise, and training on noise improved robustness to small perturbations.

In the introduction we referred to a question from Szegedy et al. (2014) about why we find errors so close to our test points while the test error itself is so low. We can now suggest an answer: despite what our low-dimensional visual intuition may lead us to believe, these errors are not in fact unnaturally close given the error rates we observe in noise. There is a sense, then, in which we simply haven't reduced the test error enough to expect to have removed most nearby errors.

While we focused on the Gaussian distribution, similar conclusions can be made about other distributions. In general, in high dimensions, the $\epsilon$-boundary measure of a typical set is large even when its volume is small, and this observation does not depend on anything specific about the Gaussian distribution. The Gaussian distribution is a special case in that we can easily prove that *all* sets will have large $\epsilon$-boundary measure. Mahloujifar et al. (2018) proved a similar theorem for a larger class of distributions. For other data distributions not *every* set has large $\epsilon$-boundary measure, but under some additional assumptions it still holds that *most* sets do. An investigation of this relationship on the MNIST distribution can be found in Gilmer et al. (2018b, Appendix G).

We believe it would be beneficial for the adversarial defense literature to start reporting generalization in noisy image distributions, such as the common corruption benchmark introduced in Hendrycks & Dieterich (2018), rather than the current practice of *only* reporting empirical estimates of adversarial robustness. There are several reasons for this recommendation.

1. Measuring test error in noise is significantly easier than measuring adversarial robustness — computing adversarial robustness perfectly requires solving an NP-hard problem for every point in the test set (Katz et al., 2017). Since Szegedy et al. (2014), hundreds of adversarial defense papers have been published. To our knowledge, only one (Madry et al., 2017) has reported robustness numbers which were confirmed by a third party. We believe the difficulty of measuring robustness under the usual definition has contributed to this unproductive situation.

2. Measuring test error in noise would also allow us to determine whether or not these methods improve robustness in a trivial way, such as how the robust MNIST model learned to threshold the input, or whether they have actually succeeded in improving generalization outside the natural data distribution.

3. All of the failed defense strategies we examined failed to improve generalization in noise. For this reason, we should be highly skeptical of defense strategies that only claim improved $l_p$-robustness but do not demonstrate robustness in more general settings.

4. Finally, if the goal is improving the security of our models in adversarial settings, errors in the presence of noise are already indicative that our models are not secure. Until our models are perfectly robust in the presence of average-case corruptions, they will not be robust in worst-case settings. The usefulness of $l_p$-robustness in realistic threat models is limited when attackers are not constrained to making small modifications.

The interest in measuring $l_p$ robustness arose from a sense of surprise that errors could be found so close to correctly classified points. But from the perspective described in this paper, the phenomenon is less surprising. Statistical classifiers make a large number of errors outside the data on which they were trained, and small adversarial perturbations are simply the nearest ones.

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

| $\sigma$ | 0.00625 | 0.0125 | 0.025 | 0.075 | 0.15 | 0.25 |
|---|---|---|---|---|---|---|
| Training Accuracy | 100% | 100% | 100% | 100% | 99.9% | 99.4% |
| Test Accuracy | 96.0% | 95.5% | 94.8% | 90.4% | 77.5% | 62.2% |

Table 2: Wide ResNet-28-10 (Zagoruyko & Komodakis, 2016) trained and tested on CIFAR-10 with Gaussian noise with standard deviation $\sigma$.

| $\sigma$ | 0 | 0.1 | 0.2 | 0.4 | 0.6 | 0.8 |
|---|---|---|---|---|---|---|
| Clean Training Accuracy | 91.5% | 90.8% | 89.9% | 87.7% | 86.1% | 84.6% |
| Clean Test Accuracy | 75.9% | 75.5% | 75.2% | 74.2% | 73.3% | 72.4% |
| Noisy Training Accuracy | − | 89.0% | 85.7% | 78.3% | 71.7% | 65.2% |
| Noisy Test Accuracy | − | 73.9% | 70.9% | 65.2% | 59.7% | 54.0% |

Table 3: The models from Section 1 trained and tested on ImageNet with Gaussian noise with standard deviation $\sigma$; the column labeled 0 refers to a model trained only on clean images.

## A    TRAINING DETAILS

**Models trained on CIFAR-10.** We trained the Wide-ResNet-28-10 model (Zagoruyko & Komodakis, 2016) using standard data augmentation of flips, horizontal shifts and crops in addition to Gaussian noise independently sampled for each image in every minibatch. The models were trained with the open-source code by Cubuk et al. (2018) for 200 epochs, using the same hyperparameters which we summarize here: a weight decay of 5e-4, learning rate of 0.1, batch size of 128. The learning rate was decayed by a factor of 0.2 at epochs 60, 120, 160.

**Models trained on ImageNet.** The ResNet-50 model (He et al., 2016) was trained with a learning rate of 1.6, batch size of 4096, and weight decay of 1e-4. During training, random crops and horizontal flips were used, in addition to the Gaussian noise independently sampled for each image in every minibatch. The models were trained for 90 epochs, where the learning rate was decayed by a factor of 0.1 at epochs 30, 60, and 80. Learning rate was linearly increased from 0 to the value of 1.6 over the first 5 epochs.

## B    NOISE ATTACK DETAILS

Here we provide more detail for the noise distributions considered in Section 5. The stAdv attack defines a flow field over the pixels of the image and shifts the pixels according to this flow. The field is parameterized by a latent $Z$. When we measure accuracy against our randomized variant of this attack, we randomly sample $Z$ from a multivariate Gaussian distribution with standard deviation $\sigma$. To implement this attack we used the open sourced code from Xiao et al. (2018). PCA-100 noise first samples noise from a Gaussian distribution $\mathcal{N}(0, \sigma)$, and then projects this noise onto the first 100 PCA components of the data. For ImageNet, the input dimension is too large to perform a PCA decomposition on the entire dataset. So we first perform a PCA decomposition on 30x30x1 patches taken from different color channels of the data. To general the noise we first sample from a 900 dimensional Gaussian, then project this into the basis spanned by the top 100 PCA components, then finally tile this projects to the full 299x299 dimension of the input. Each color channel is constructed independently in this fashion.

## C    TRAINING AND TESTING ON GAUSSIAN NOISE

In Section 5, we mentioned that it is not trivial to learn the distribution of noisy images simply by augmenting the training data distribution. In Tables 2 and 3 we present more information about the performance of the models we trained and tested on various scales of Gaussian noise.

|  | Clean Accuracy | Pepper $p = 0.2$ Accuracy | Gaussian $\sigma = 0.3$ Accuracy | stAdv $\sigma = 1.0$ Accuracy | PCA-100 $\sigma = 0.3$ Accuracy |
|---|---|---|---|---|---|
| Model |  |  |  |  |  |
| Clean | 99.2% | 81.4% | 96.9% | 89.5% | 63.3% |
| Adv | 98.4% | 27.5% | 78.2% | 93.2% | 47.1% |

Table 4: The performance of ordinarily and adversarially trained MNIST models on various noise distributions.

## D   RESULTS ON MNIST

MNIST is a special case when it comes to the relationship between small adversarial perturbations and generalization in noise. Indeed prior has already observed that an MNIST model can trivially become robust to small $l_\infty$ perturbations by learning to threshold the input (Schmidt et al., 2018), and observed that the model from Madry et al. (2017) indeed seems to do this. When we investigated this model in different noise distributions we found it generalizes worse than a naturally trained model, results are shown in Table 4. Given that it is possible for a defense to overfit to a particular $l_p$ metric, future work would be strengthened by demonstrating improved generalization outside the natural data distribution.

## E   THE GAUSSIAN ISOPERIMETRIC INEQUALITY

Here we will discuss the Gaussian isoperimetric inequality more thoroughly than we did in the text. We will present some of the geometric intuition behind the theorem, and in the end we will show how the version quoted in the text follows from the form in which the inequality is usually stated.

The historically earliest version of the isoperimetric inequality, and probably the easiest to understand, is about areas of subsets of the plane and has nothing to do with Gaussians at all. It is concerned with the following problem: among all measurable subsets of the plane with area $A$, which ones have the smallest possible perimeter?[1] One picture to keep in mind is to imagine that you are required to fence off some region of the plane with area $A$ and you would like to use as little fence as possible. The isoperimetric inequality says that the sets which are most "efficient" in this sense are balls.

Some care needs to be taken with the definition of the word "perimeter" here — what do we mean by the perimeter of some arbitrary subset of $\mathbb{R}^2$? The definition that we will use involves the concept of the $\epsilon$-boundary measure we discussed in the text. For any set $E$ and any $\epsilon > 0$, recall that we defined the $\epsilon$-extension of $E$, written $E_\epsilon$, to be the set of all points which are within $\epsilon$ of a point in $E$; writing $A(E)$ for the area of $E$, we then define the perimeter of $E$ to be

$$\text{surf}(E) := \liminf_{\epsilon \to 0} \frac{1}{\epsilon} \left( A(E_\epsilon) - A(E) \right).$$

A good way to convince yourself that this is reasonable is to notice that, for small $\epsilon$, $E_\epsilon - E$ looks like a small band around the perimeter of $E$ with width $\epsilon$. The isoperimetric inequality can then be formally expressed as giving a bound on the quantity inside the limit in terms of what it would be for a ball. (This is slightly stronger than just bounding the perimeter, that is, bounding the limit itself, but this stronger version is still true.) That is, for any measurable set $E \subseteq \mathbb{R}^2$,

$$\frac{1}{\epsilon}(A(E_\epsilon) - A(E)) \geq 2\sqrt{\pi A(E)} + \epsilon\pi.$$

It is a good exercise to check that we have equality here when $E$ is a ball.

There are many generalizations of the isoperimetric inequality. For example, balls are also the subsets in $\mathbb{R}^n$ which have minimal surface area for a given fixed volume, and the corresponding set on the surface of a sphere is a "spherical cap," the set of points inside a circle drawn on the surface of the sphere. The version we are most concerned with in this paper is the generalization to a Gaussian distribution. Rather than trying to relate the volume of $E$ to the volume of $E_\epsilon$, the Gaussian

---

[1]The name "isoperimetric" comes from a different, but completely equivalent, way of stating the question: among all sets with the same fixed perimeter, which ones have the largest possible area?

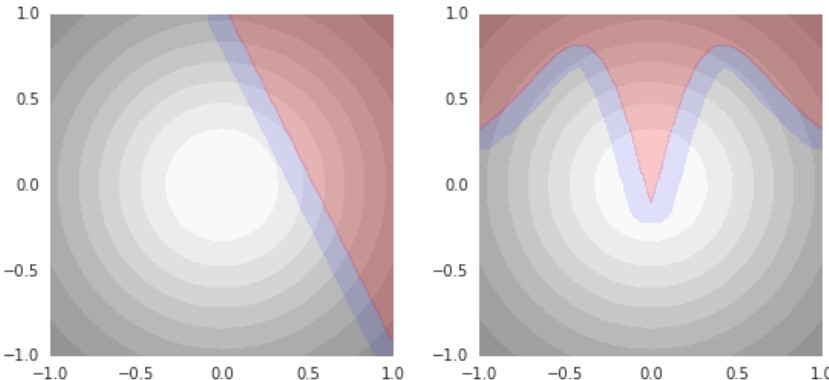

Figure 6: The Gaussian isoperimetric inequality relates the amount of probability mass contained in a set $E$ to the amount contained in its $\epsilon$-extension $E_\epsilon$. A sample from the Gaussian is equally likely to land in the pink set on the left or the pink set on the right, but the set on the right has a larger $\epsilon$-extension. The Gaussian isoperimetric inequality says that the sets with the smallest possible $\epsilon$-extensions are half spaces.

isoperimetric inequality is about the relationship between the *probability* that a random sample from the Gaussian distribution lands in $E$ or $E_\epsilon$. Other than this, though, the question we are trying to answer is the same: for a given probability $p$, among all sets $E$ for which the probability of landing in $E$ is $p$, when is the probability of landing in $E_\epsilon$ as small as possible?

The Gaussian isoperimetric inequality says that the sets that do this are half spaces. (See Figure 6.) Just as we did in the plane, it is convenient to express this as a bound on the probability of landing in $E_\epsilon$ for an arbitrary measurable set $E$. This can be stated as follows:

**Theorem.** *Consider the standard normal distribution $q$ on $\mathbb{R}^n$, and let $E$ be a measurable subset of $\mathbb{R}^n$. Write*

$$\Phi(t) = \frac{1}{\sqrt{2\pi}} \int_{-\infty}^{t} \exp(x^2/2)dx,$$

*the cdf of the one-variable standard normal distribution.*

*For a measurable subset $E \subseteq \mathbb{R}^n$, write $\alpha(E) = \Phi^{-1}(\mathbb{P}_{x\sim q}[x \in E])$. Then for any $\epsilon \geq 0$,*

$$\mathbb{P}_{x\sim q}[x \in E_\epsilon] \geq \Phi(\alpha(E) + \epsilon).$$

The version we stated in the text involved $\epsilon_q^*(E)$, the median distance from a random sample from $q$ to the closest point in $E$. This is the same as the smallest $\epsilon$ for which $\mathbb{P}_{x\sim q}[x \in E_\epsilon] = \frac{1}{2}$. So, when $\epsilon = \epsilon_q^*(E)$, the left-hand side of the Gaussian isoperimetric inequality is $\frac{1}{2}$, giving us that $\Phi(\alpha + \epsilon_q^*(E)) \leq \frac{1}{2}$.

Since $\Phi^{-1}$ is a strictly increasing function, applying it to both sides preserves the direction of this inequality. But $\Phi^{-1}(\frac{1}{2}) = 0$, so we in fact have that $\epsilon_q^*(E) \leq -\alpha$, which is the statement we wanted.

## F    VISUALIZING THE OPTIMAL CURVES

The optimal bound according to the isoperimetric inequality gives surprisingly strong bounds in terms of the existence of worst-case $l_2$ perturbations and error rates in Gaussian noise. In Figure 7 we plot the optimal curves for various values of $\sigma$, visualize images sampled from $x + N(0,\sigma)$, and visualize images at various $l_2$ distance from the unperturbed clean image. Even for very large noise ($\sigma = .6$), test error needs to be less than $10^{-15}$ in order to have worst-case perturbations be larger than 5.0. In order to visualize worst-case perturbations at varying $l_2$ distances, we visualize an image that minimizes similarity according to the SSIM metric (Wang & Bovik, 2009). These images are found by performing gradient descent to minimize the SSIM metric subject to the containt that $||x - x_{adv}||_2 < \epsilon$.

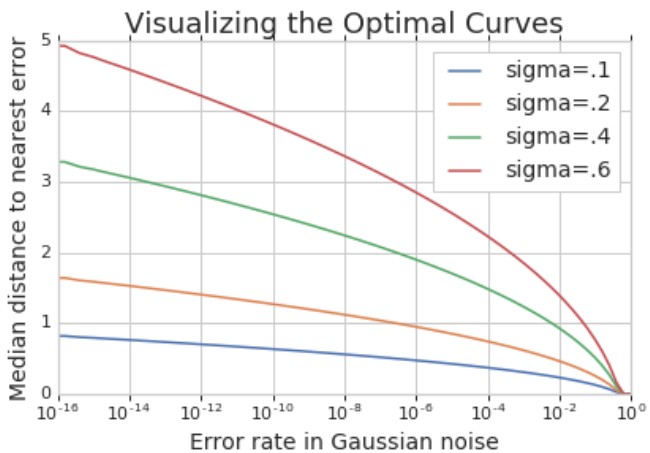

**Random Gaussian Perturbations of the Clean Image**

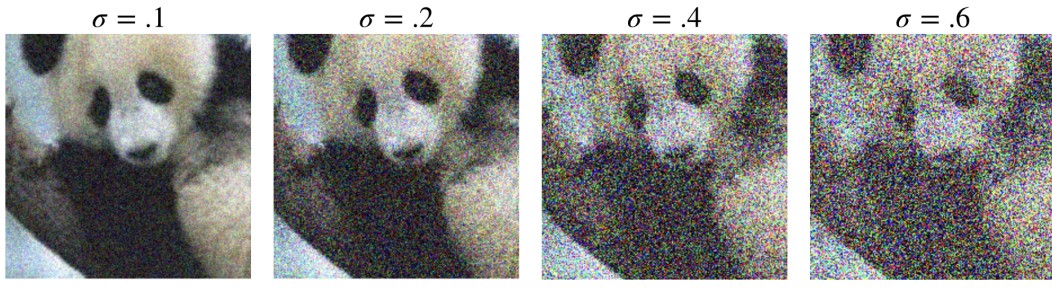

**Small L2 Perturbations of the Clean Image**

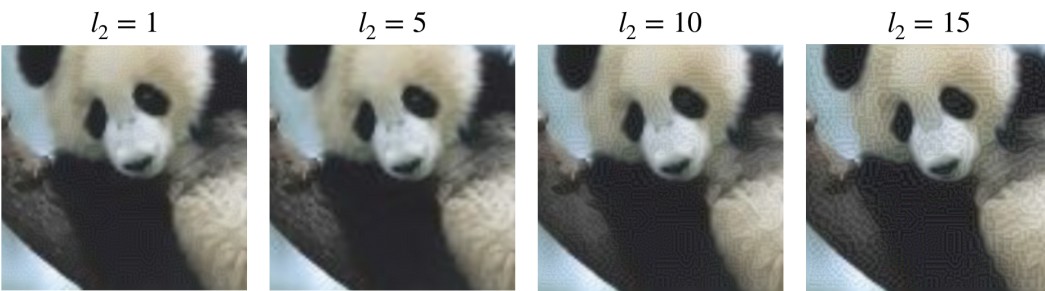

Figure 7: **Top:** The optimal curves on Imagenet for different values of $\sigma$. **Middle:** Visualizing different coordinates of the optimal curves. First, random samples from $x + N(0, \sigma I)$ for different values of $\sigma$. **Bottom:** Images at different $l_2$ distances from the unperturbed clean image. Each image visualized is the image at the given $l_2$ distance which minimizes visual similarity according to the SSIM metric. Note that images at $l_2 < 5$ have almost no perceptible change from the clean image despite the fact that SSIM visual similarity is minimized.

## G CHURCH WINDOW PLOTS

In this section we include many more visualizations of the sorts of church window plots we discussed briefly in Section 4. We will show an ordinarily trained model's predictions on several different slices through the same CIFAR test point which illustrate different aspects of the story told in this paper. These images are best viewed in color.

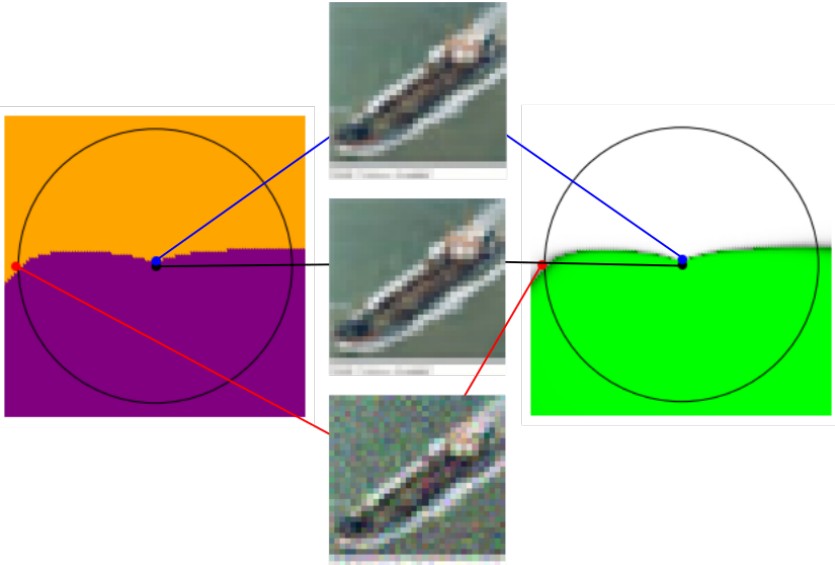

Figure 8: A slice through a clean test point (black, center image), the closest error found using PGD (blue, top image), and a random error found using Gaussian noise (red, bottom image). For this visualization, and all others in this section involving Gaussian noise, we used noise with $\sigma = 0.05$, at which the error rate was about 1.7%. In all of these images, the black circle indicates the distance at which the typical such Gaussian sample will lie. The plot on the right shows the probability that the model assigned to its chosen class. Green indicates a correct prediction, gray or white is an incorrect prediction, and brighter means more confident.

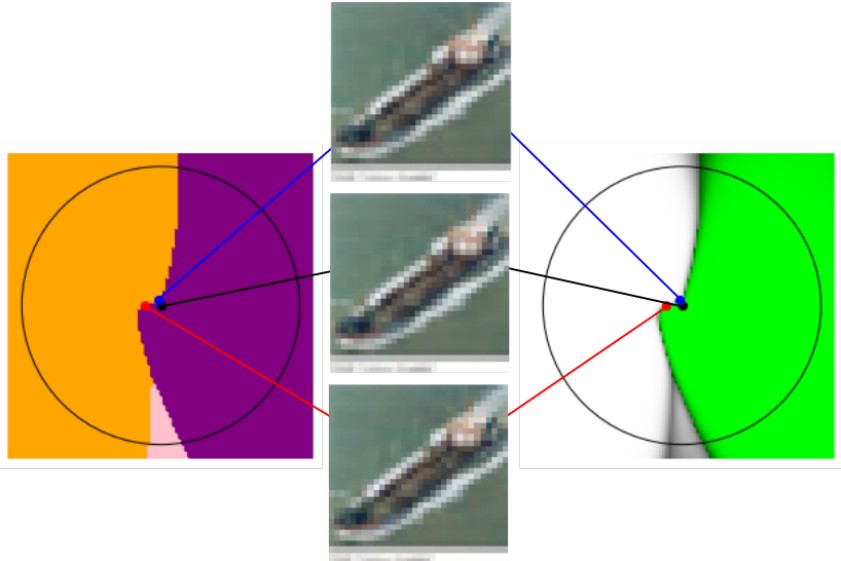

Figure 9: A slice through a clean test point (black, center image), the closest error found using PGD (blue, top image), and the average of a large number of errors randomly found using Gaussian noise (red, bottom image). The distance from the clean image to the PGD error was 0.12, and the distance from the clean image to the averaged error was 0.33. The clean image is assigned the correct class with probability 99.9995% and the average and PGD errors are assigned the incorrect class with probabilities 55.3% and 61.4% respectively. However, it is clear from this image that moving even a small amount into the orange region will increase these latter numbers significantly. For example, the probability assigned to the PGD error can be increased to 99% by moving it further from the clean image in the same direction by a distance of 0.07.

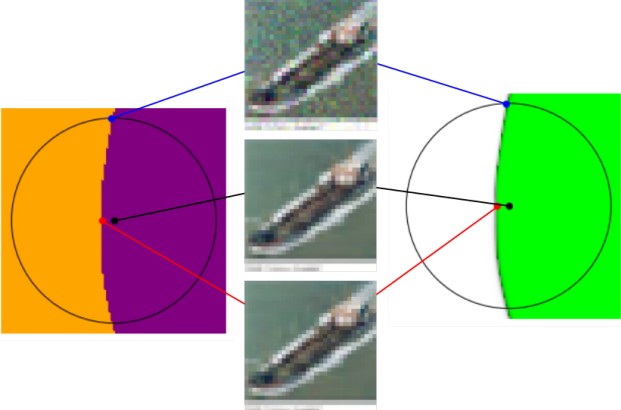

Figure 10: A slice through a clean test point (black, center image), a random error found using Gaussian noise (blue, top image), and the average of a large number of errors randomly found using Gaussian noise (red, bottom image).

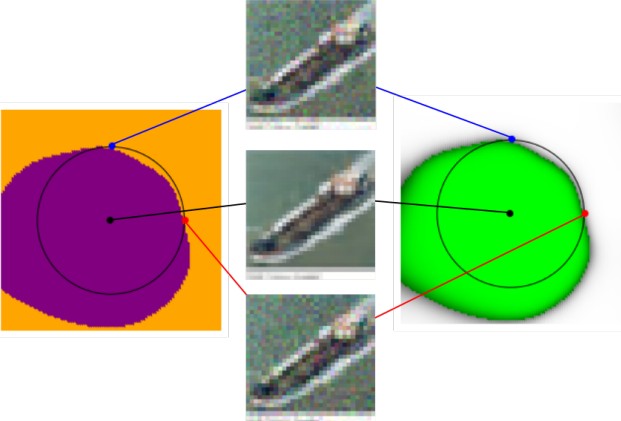

Figure 11: A slice through a clean test point (black, center image) and two random errors found using Gaussian noise (blue and red, top and bottom images). Note that both random errors lie very close to the decision boundary, and in this slice the decision boundary does not appear to come close to the clean image.

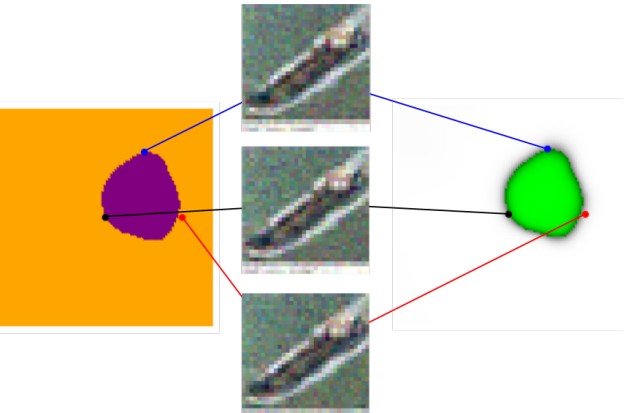

Figure 12: A slice through three random errors found using Gaussian noise. (Note, in particular, that the black point in this visualization does not correspond to the clean image.)

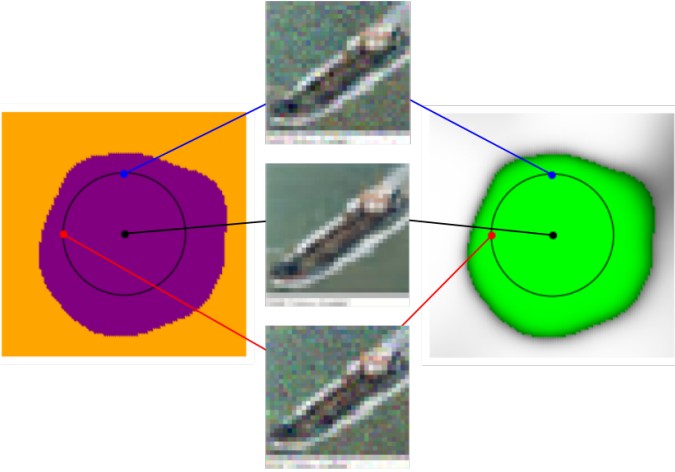

Figure 13: A completely random slice through the clean image.

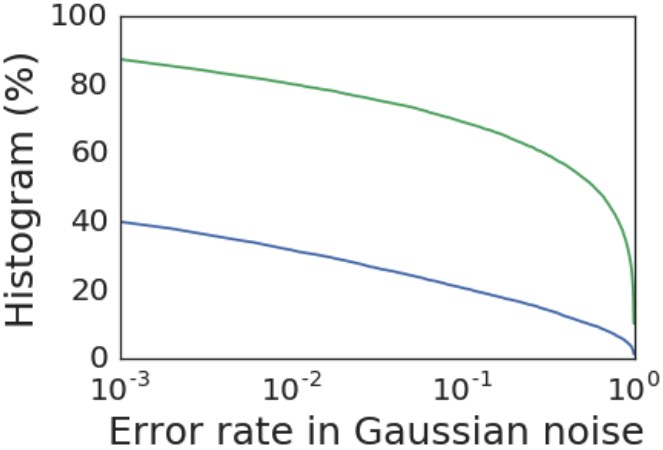

Figure 15: The cdf of the error rates in noise for images in the test set. The blue curve corresponds to a model trained and tested on noise with $\sigma = 0.1$, and the green curve is for a model trained and tested at $\sigma = 0.3$. For example, the left most point on the blue curve indicates that about 40% of test images had an error rate of at least $10^{-3}$.

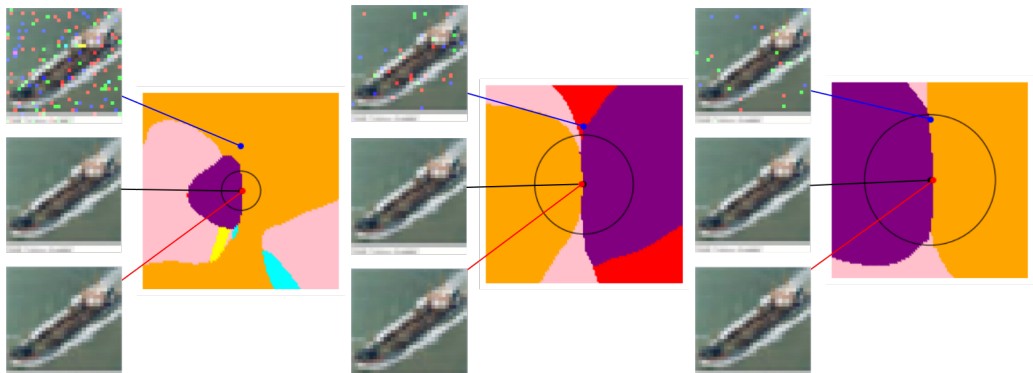

Figure 14: Some visualizations of the same phenomenon, but using the "pepper noise" discussed in Section 5 rather than Gaussian noise. In all of these visualizations, we see the slice through the clean image (black, center image), the same PGD error as above (red, bottom image), and a random error found using pepper noise (blue, top image). In the visualization on the left, we used an amount of noise that places the noisy image further from the clean image than in the Gaussian cases we considered above. In the visualization in the center, we selected a noisy image which was assigned to neither the correct class nor the class of the PGD error. In the visualization on the right, we selected a noisy image which was assigned to the same class as the PGD error.

## H    THE DISTRIBUTION OF ERROR RATES IN NOISE

Using some of the models that were trained on noise, we computed, for each image in the CIFAR test set, the probably that a random Gaussian perturbation will be misclassified. A histogram is shown in Figure 15. Note that, even though these models were trained on noise, there are still many errors around most images in the test set. While it would have been possible for the reduced performance in noise to be due to only a few test points, we see clearly that this is not the case.

# I  A COLLECTION OF MODEL ERRORS

In this section we first show a collection of iid test errors for the ResNet-50 model on the ImageNet validation set. We also visualize the severity of the different noise distributions considered in this work, along with model errors found by random sampling in these distributions.

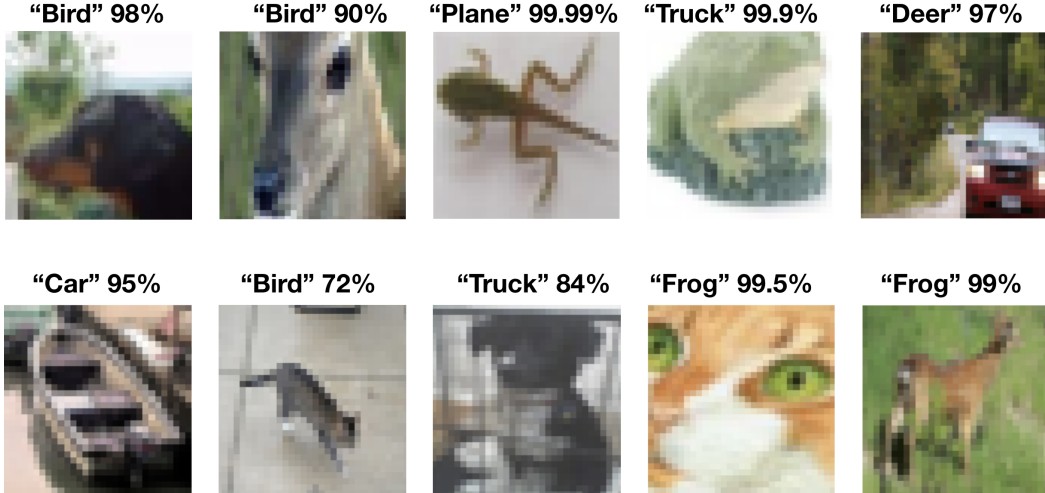

Figure 16: A collection of adversarially chosen model errors. These errors appeared in the ImageNet validation set. Despite the high accuracy of the model there remain plenty of errors in the test set that a human would not make.

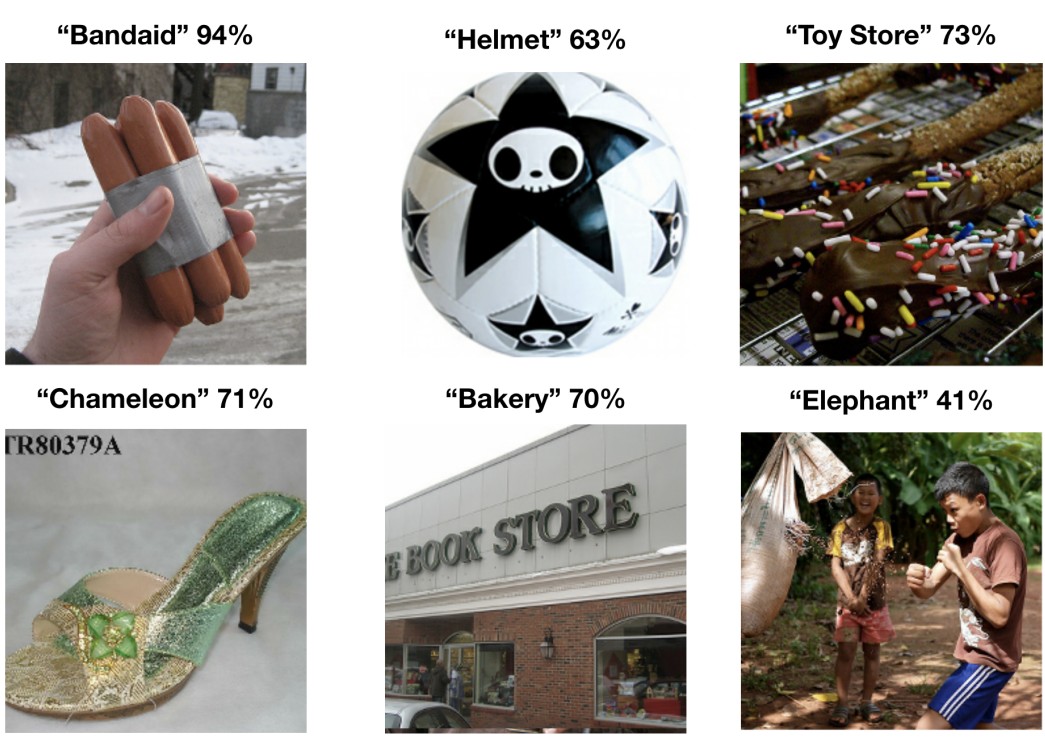

Figure 17: A collection of adversarially chosen model errors. These errors appeared in the ImageNet validation set. Despite the high accuracy of the model there remain plenty of errors in the test set that a human would not make.

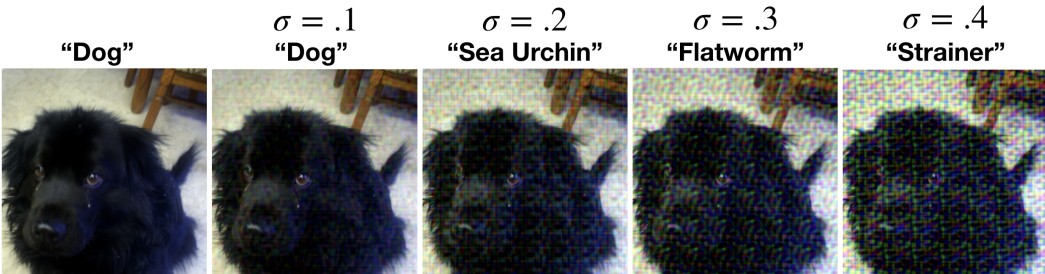

Figure 18: Visualizing the severity of PCA noise, along with model errors found in this noise distribution.

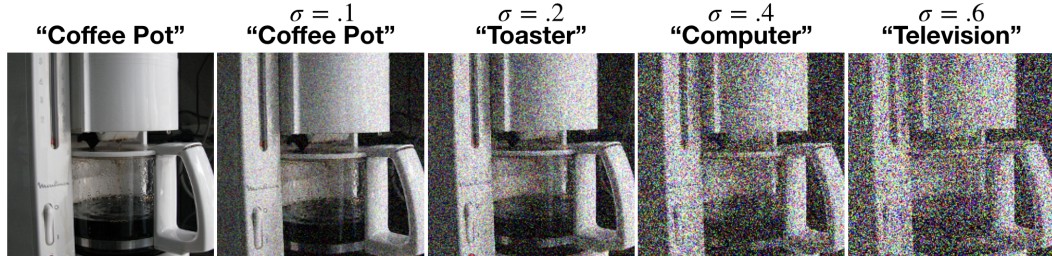

Figure 19: Visualizing the severity of Gaussian noise, along with model errors found in this noise distribution. Note the model shown here was trained at noise level $\sigma = .6$.

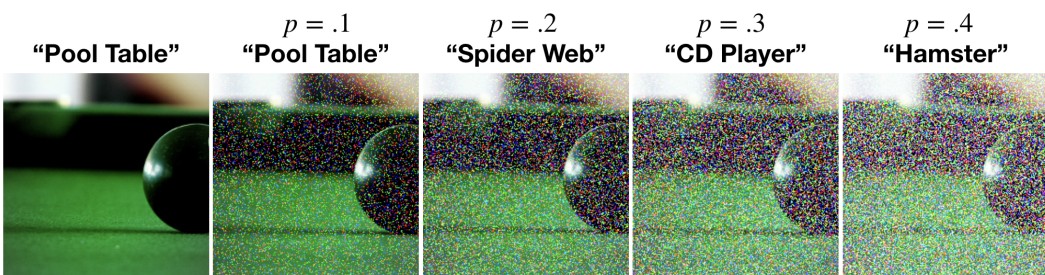

Figure 20: Visualizing the severity of pepper noise.

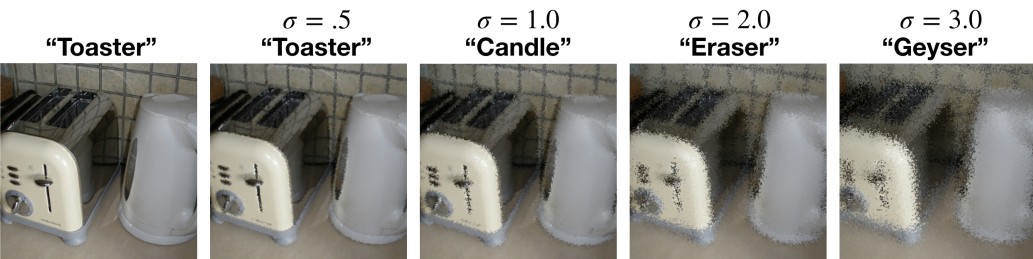

Figure 21: Visualizing the severity of the randomized stAdv attack.

