# OpenReview forum: "Adversarial Examples Are a Natural Consequence of Test Error in Noise"
_ICLR.cc/2019/Conference_

### Official Review · AnonReviewer2 · 2018-11-01
**A connection between training with noisy images and the adversarial training is not convincing**

**Rating:** 4
**Confidence:** 3

**Review:**

The paper suggests a connection between training with noisy images and the adversarial training. The observation is original to me. It has several cons.

1. The paper is hard to follow because of  too many vague descriptions and unnecessary contrast clauses. Here are some.
 1) The title of section 4: ERRORS IN NOISE IMPLY ADVERSARIAL EXAMPLES FOR NOISY IMAGE.
 2) "The discussion of high-dimensional geometry suggests that adversarial examples may actually not be in contradiction to high generalization performance. Indeed, high generalization performance does not mean perfect generalization" The uncertain tone "may not" and the vague statement "does not mean perfect generalization" make readers hard to get the solid understanding about what the paper tries to say.
 3)  "Adversarial training on small perturbations on CIFAR-10 also improved generalization in noise, and training on noise improved robustness to small perturbations."
Many other sentences like the above make the paper not technically sound.

2. It is problematic that the paper uses Euclidean l2 distance to measure the error set and its surface as it is believed that the dataset lives on low-dimensional manifold. Moreover, the adversarial examples often constructed by moving the legal images towards a specific direction rather than adding the Gaussian isotropic noise.
3. The advocates that using test error in noise as a measure of adversarial robustness  is misleading  as test error in noise has a large number of different combinations: noise type, noise amplitude. One may find one type of test error in noise coinciding with adversarial robustness but  in general it is not a good measure for adversarial robustness because of its varying nature.
4. Several terms are referred without definitions: errors in noise,  adversarial robustness. From the definition of E_epsilon, it should include the interior of E.

---

### Official Review · AnonReviewer1 · 2018-11-02
**Interesting but algorithms too similar to previous work**

**Rating:** 5
**Confidence:** 3

**Review:**

This paper propose an alternative view for adversarial examples in high dimension spaces by considering the "error rate" in a Gaussian distribution centered at each test point. However, as mentioned in the related work, adversarial examples through the lens of isoperimetric inequality is not new to this paper; the implication of adversarial sensitivity by error rate in the test-sample-centered-Gaussian in general non-linear case is rather weak; and the empirical results does not show advantage over simple adversarial training against lp constrained adversarial attacks.

Here are some more detailed comments and feedbacks:

The clarity could be improved by making clear use of notations and define some key terms explicitly:

1. For example, the error rate sometimes refer to the test error, sometimes refer to the error rate under a special distribution centered at a particular test example.

2. Similarly, the distribution q sometimes refer to the original (unknown) input data distribution, but the same notation is also used to refer to this Gaussian distribution centered on each test example. Although the paper says that "q need not be restricted to the distribution from which the training set was sampled", it could potentially confuse the reader less if there is a symbol for the "usual" test error and a different one for this test-error-under-Gaussian-centered-at-a-particular-test-example.

3. It would be good if the paper could make a formal definition of the problem being studied and explicitly specify the assumptions on the existence of a deterministic target function (concept) and explicitly define the error set E.

4. It seems to assume the input distribution is continuous everywhere but not stated. If for example, the original data is supported on disconnected manifolds separated with low density or even zero-density margins, then the Gaussian distribution centered on test example argument will need to be modified to talk about the intersection of the Gaussian with the data manifold instead. If the paper does decide to make this kind of assumption, some empirical study on the data to verify the fidelity of the assumptions would be great.

5. The paper does not mention how measurement against the error set E. Under the original data distribution, it is natural to measure the error rate with the provided training or test data with labels. However, under each newly formed Gaussian distribution centered at each test point, the labels for the newly sampled examples from this Gaussian is unknown, and since there is no "ground truth classifier" for MNIST or CIFAR10 available, it seems impossible to "calculate" the true label for those samples, which are needed to calculate the error rate. It is not very clear from the paper how this issue is solved. I'm guess it uses the label from the Gaussian center for all the samples from the Gaussian. While this might be reasonable assumption for Gaussian with tiny variances, it is less clear how reasonable it is for the large variance Gaussian distributions considered in the paper. If this assumption is made, please state it explicitly and empirically or theoretically study how reasonable this assumption is in the regime of variances considered in this paper. If my guessing is wrong, please also explicitly what approach is used to get around this issue.

The followings are some feedbacks on the contents and ideas of the paper:

6. I think one sentence in the text summarize a large part of the paper very well: "to measure adversarial robustness is to ask whether or not there are any errors in the linf ball, ... and to measure test error in noise is to measure the volume of the error set in the defined noise distribution". However, this looks like a rather roundabout approach to attack another problem (measuring volume of an unknown set in a very high dimension space) in order to solve the original problem, while the implication is rather weak (a precise implication can only be obtained for linear separating hyper-plane, while for non-linear classifiers it is much less clear).

Note while exact adversarial robustness is NP-hard, volume estimation in high dimension is not easy (if not harder). For cifar-10, the inputs are in dimension 32x32x3 = 3072, the 1,000 samples used in the paper to estimate the volume of a set in this high dimension seem to be quite inaccurate. I would appreciate if variances could be reported in those studies to show the confidence of the estimations. For imagenets, the inputs are in even larger spaces.

Given the difficulty (in terms of sample complexity) to estimate the "error-in-noise", it might not be very surprising that the noise augmentation does not show advantages to lp constrained adversarial attacks (comparing to adversarial training).

7. In the conclusion, the paper states "we proved a fundamental relationship between generalization in noisy image distributions and the existence of small adversarial perturbations". I believe a formal proof is only given to the case of existence of small adversarial perturbations to examples to the "noisy examples" from the Gaussian distribution, and in this case, it is a rather direct corollary from the Gaussian Isoperimetric Inequality. For the more "practical case" (in the sense that is more related to the usual notion of adversarial examples) of existence of small adversarial examples, it seems only the case of the linear classifier is formally discussed.

Moreover, I'm a little bit worried some important pieces might be lost and create potentially misleading or seemly strong conclusion. Maybe it would be helpful if a concrete example could be given in the paper that shows the full path from the error-in-noise to existence of adversarial example, by showing all the constants involved. I'm a bit confused here because (in order for the isoperimetric inequality to have favorable bounds?) the Gaussian distributions used in error-in-noise seem to have rather large variance. As mentioned in the paper, the majority of the mass in the Gaussian distribution considered will be in a thin sphere of radius sigma * sqrt(n) centered at the test example x. If sigma = 0.1 and dimension n = 3072 (cifar-10), then the radius is around 5.5 (in l2 distance) which is probably quite far from the test point x (is it?). It is then less clear how "a large majority of this thin sphere far away from x is epsilon close to the error set" could tightly imply properties of adversarial robustness of x itself in its close vicinity. Maybe a specific example with all the numerical constants spelled out would help illustrate this.

In summary, I think this paper takes an interesting but roundabout perspective to adversarial robustness, and the implication is weak in the non-linear case. (Potentially because of the weak implication), the suggested approach for defenses by augmenting with noises does not show advantage over adversarial training.

---

> ### Author Response · Authors · 2018-11-23
> **Thank you for the detailed review, if you have time can you answer some further questions so we can improve upon this work?**
>
> It would help us improve the paper to get some additional clarity on the parts you find confusing or not interesting. Would you mind responding to a few of our questions?
>
> "7. In the conclusion, the paper states "we proved a fundamental relationship between generalization in noisy image distributions and the existence of small adversarial perturbations". I believe a formal proof is only given to the case of existence of small adversarial perturbations to examples to the "noisy examples" from the Gaussian distribution, and in this case, it is a rather direct corollary from the Gaussian Isoperimetric Inequality. For the more "practical case" (in the sense that is more related to the usual notion of adversarial examples) of existence of small adversarial examples, it seems only the case of the linear classifier is formally discussed."
>
> You are absolutely right that the isoperimetric inequality does not apply to small perturbations of the clean image. However for the adversarial examples shown in Figure 2 it directly applies. Is there a reason you don't find the existence of the adversarial examples in Figure 2 interesting? It's still a small perturbation of a correctly classified image.
>
> "Moreover, I'm a little bit worried some important pieces might be lost and create potentially misleading or seemly strong conclusion. Maybe it would be helpful if a concrete example could be given in the paper that shows the full path from the error-in-noise to existence of adversarial example, by showing all the constants involved. I'm a bit confused here because (in order for the isoperimetric inequality to have favorable bounds?) the Gaussian distributions used in error-in-noise seem to have rather large variance."
>
> Let us walk you through the adversarial example shown in the left side of Figure 5, because it should help illustrate the full argument of the paper. Figure 5 shows a 2d-slice of the decision boundary through 3 images, the clean image, an adversarial example found via PGD, and a randomly found error in noise at sigma=.08. The PGD adversarial example has l2 distance of .274 from the clean image, the error in noise as l2 distance of 31.4. Note that despite the noisy images having an l2 distance of 31.4, it still is easily identified as a panda. We found that .1% of the random gaussian perturbations are misclassified by this model. Visually, this means that .1% of the sphere at radius 31.4 is misclassified, although in this 2d slice it looks like most of the sphere is misclassified. The isoperimetric inequality then implies that the median distance from a noisy image to an error is at most -.08 * phi^{-1}(.1) = .24. This means that most random Gaussian perturbations, although correctly classified must be very close to the error set (50% are within distance .24).
>
> Now you are correct that this doesn’t say anything about the clean image, however if we make a linearity assumption (which the CIFAR-10 and Imagenet models all satisfied empirically, as shown in Figures 3 and 4), then errors in noise will imply small perturbations of the clean image and the relationship between the two is the same bound implied by the isoperimetric inequality. This helps explain why adversarial training on small perturbations reduced test error in large random noise, by pushing the decision boundary away the errors are both farther away from the clean image and the error rates in noise decreased. This is one of the reasons we recommend defense papers consider evaluating on more general noise distributions, at least in addition to the standard practice of evaluating small lp adversarial examples. The theory suggests that defenses which are truly making progress on this problem, say by enforcing a better model prior for images, should also help improve generalization in the presence of more general image corruptions.
>
> For most security settings of interest, the Gaussian isoperimetric inequality does imply that achieving perfectly robust models requires reducing test error in noise to essentially 0 (note for many settings of interest attackers will not be restricted to making only small perturbations of the clean image). For example: defending against worst case perturbations of distance .6 requires the error rate at sigma=.1 to be less than 10^(-10). Now we think this is significant because for most security settings of interest, an attacker will not be restricted to small perturbations of the clean image. Furthermore, the small perturbations of noisy images expose the same sort of blind spots that perturbations of the clean images do. Is there a reason you find small perturbations of noisy images less surprising or interesting than small perturbations of the clean image?

---

> > ### Author Response · Authors · 2018-11-23
> > **part 2**
> >
> > Can you clarify what you mean by algorithms too similar to prior work? We aren’t introducing any algorithms in this work, we are presenting a theoretical understanding of the adversarial example phenomenon. While the isoperimetric inequality has been studied in some prior works, we are the first work to empirically compare the robustness of vision models with respect to the bounds applied by the isoperimetric inequality. The reason we considered models trained in additive noise was to confirm the connection between small worst-case perturbations and large average case perturbations, rather than propose a new defense. The fact that standard Gaussian data augmentation improved adversarial robustness, and that adversarial training improved robustness in Gaussian noise solidifies the connection we are making.
> >
> > Thanks again for your comments. If you have further questions, or comments on the paper please don’t hesitate to ask. Your comments so far have already helped improve this work.

---

### Official Review · AnonReviewer3 · 2018-11-03
**not very interesting conclusion**

**Rating:** 4
**Confidence:** 4

**Review:**

The paper tries to make a connection between the functionality of Gaussian noise to adversarial examples. It shows that data augmentation with added Gaussian noise could also improve the model robustness.
Meanwhile, it shows that in a high dimensional space, even with a small (error) set, its bounding ball of \epsilon l_p distance could be large. It explains why even with a small test error, the model could still be vulnerable to adversarial examples.

Although the paper has some good intuitions and some nice experiments, I find the main conclusion of this paper not very interesting.
Although I partially buy the second point about the high dimensional geometry, this is an obvious observation and does not give rise to much meaningful result for the future work. It would be more interesting to see the different geometry structure of robust versus not robust models.

Meanwhile, though a formal definition of error set is not presented in the paper, it seems the authors are simply dividing the data space to the set where the model gives a correct label, and the “error set” the other way around. However, since the paper is considering a data distribution (q) rather than a dataset, the separation could be more complicated than that. For instance, a noise image should also in your data space, but does it belong to an error set or not? It isn’t necessarily attached to any labels. Or does your model only consider meaningful images? But what if adding noise simply get you out of the space? It’s better to make this concept clearer.


It is not a surprising result that there is one randomly chosen direction mimicking the performance of adversarial examples as in Figure 2. By “carefully crafted imperceptible noise”, I assume it means choosing one random sample that will change the model output the most. This is exactly a way of choosing an adversarial example. Since even in high dimensional space, out of a lot of random vectors , one could approximate a target (adversarial) direction.

Similar explanations also apply to the training with error part. How much more data do you use for the data augmentation? If you use much more data with Gaussian noise than what your use for adversarial training, it is not surprising at all to get a more robust network, with a similar argument as above.


minor issue: Section 3 should not be an isolated section.

---

> ### Author Response · Authors · 2018-11-29
> **Can you clarify why you find the conclusion not interesting**
>
> Thanks for your comments. However, we feel you haven’t addressed the main point of our paper in your review.
>
> Our results draw a close connection between the adversarial defense literature and the robustness literature [1,2,3,4]. This is important and significant because it suggests that methods useful for improving one metric (adversarial robustness) should also be useful for improving another metric (test error as measured in corrupted image distributions) and vice-versa. Previously, these two communities have largely studied model robustness independently of each other but it turns out in fact, they are effectively studying the same phenomenon through the lens of two different metrics. This connection dramatically changes how we view “adversarial examples”, most adversarial example papers do not discuss their existence or connection with respect to traditional notions of model robustness to noise.
>
> Do you not find the connection we are drawing interesting?
>
> “Understanding How Image Quality Affects Deep Neural Networks”: https://arxiv.org/pdf/1604.04004.pdf
> “Benchmarking Neural Network Robustness to Common Corruptions and Surface Variations”: https://arxiv.org/abs/1807.01697
> “Comparing deep neural networks against humans: object recognition when the signal gets weaker” https://arxiv.org/abs/1706.06969
> “An Overview of Noise-Robust Automatic Speech Recognition”: https://ieeexplore.ieee.org/document/6732927

---

### Public Comment · ~Angus_Galloway1 · 2018-10-05
**comments pt 1**

This paper makes some reasonable recommendations that can be used in works that aim to defend against adversarial perturbations, such as reporting test performance for additive noise, supplementing that for worst case perturbations. To my knowledge, this is a common benchmark in classical computer vision, but not common in adversarial examples literature. It is convincing that this practice could have partly mitigated the false sense of improved “security” due to gradient masking/overfitting specific attack methods, previously reported by Carlini and Athalye. It is similarly convincing that training on isotropic Gaussian noise can reduce sensitivity to “small” adversarial perturbations, and be competitive with adversarial training.

Unfortunately, this work makes some overly general statements.

I agree that improving adversarial robustness is complementary to generalization in the i.i.d statistical learning theory sense. However, this work seems to use “test error” and “generalization” (the difference between the empirical and expected loss) interchangeably, but the distinction between the two has major implications. The inherent trade-off between test error and adversarial robustness has been reported extensively.

https://arxiv.org/abs/1412.2309 – See discussion of spurious correlate and visual causes
https://arxiv.org/abs/1608.07690 – See Type 0, 1 adversarial examples in their taxonomy.
https://arxiv.org/abs/1804.03308 – Shows an accuracy vs robustness trade-off.
https://arxiv.org/abs/1805.12152 – The focus of their work
https://arxiv.org/abs/1808.01688 -  Characterizes the trade-off between accuracy and robustness for a variety of DNNs for image classification

The title of the present work mentions “test error in noise”. This is confusing because natural images are already noisy by default, so a possible interpretation is that we should fit the noise inherent in the dataset, which is clearly not a good idea. It is intuitive that adding isotropic Gaussian noise to the data acts as a kind of regularizer due to making the input more spherical and attenuating spurious correlates, but I am unaware of a compelling way to scale this to larger perturbations since this necessarily destroys legitimate information about the relevant variable. In any case, an important baseline would be sphering the data with ZCA, which preserves more of the relevant information for natural images, while also removing some spurious correlate and redundancy between neighboring pixels.

What is the definition of the error set E? An example is given in the context of a toy example on p2 for an input x having L2 norm less than 1, but it is unclear what this has to do with image classification? I understand that volume concentrates in a thin shell in high dimensional spaces, but I do not see how it immediately follows from the example that we are bound to have adversarial examples for high dimensional inputs in DNNs. This doesn’t tell me about the confidence with which adversarial examples may be classified (is an adversarial example corresponding to a nearly uniform softmax output really a problem?), and how the model will classify inputs that don’t cluster near any of the legitimate image volumes. Also, I think there might be an accidental overloading of terminology in reference to “typical sets”, which is a technical term in information theory, see e.g., Ch. 3 in Cover & Thomas (1991).

It is unnecessary to suggest that the only way to defend against small perturbations is to reduce the error rate in Gaussian noise, this is only one way that provably doesn’t scale. I don’t see how the Theorem regarding the Gaussian Isoperimetric Inequality precludes other approaches (e.g. preprocessing, regularization) from achieving similar results. Maybe this could be revised to say something like “it follows from … that approaches which improve adversarial robustness also improve robustness to Gaussian noise”. Also, “small” in the context of epsilon-adversarial examples was not defined, and there is no explanation as to how to choose the noise parameters, e.g., in Fig. 3, why sigma 0.1 test versus 0.4 training?

---

> ### Public Comment · ~Angus_Galloway1 · 2018-10-05
> **comments pt 2**
>
> In list item 3, it is stated that the failed defense strategies “failed to improve generalization in Gaussian noise” it is unclear what kind of improvement is expected here, and if we’re really talking about generalization or test error. It is impossible to do better in terms of test error on a test set with additive Gaussian noise, than without, because adding Gaussian noise can only destroy information. The interesting question is by how little test error degrades in the presence of this additive noise. It is difficult to draw definitive conclusions from Tables 1 and 2 because the the random error was not reported.
>
> Ultimately, while training on additive noise acts as a form of regularization on the dataset itself, it is insufficient to prevent the model from overfitting, so it’s harder to comment generally about the effect it may have on generalization gaps. Section 6 refers to a 22% gap in terms of accuracy (99 train, 77 test) on CIFAR-10 with additive Gaussian noise with sigma=0.15, this suggests that the model has excess capacity.

---

> > ### Author Response · Authors · 2018-10-05
> > **Thanks for the comments!**
> >
> > Hi, Angus,
> >
> > Thank you very much for taking the time to read the paper so closely! We’re happy that you agree with the main argument we make in the paper, and we are grateful for all the thoughtful comments. We are happy to revise the paper to clarify some of the more confusing parts you pointed to. We can also address some of them here.
> >
> > The main thrust of the paper is a connection between two metrics: adversarial robustness and test performance under additive noise. We don’t intend to make any absolute statements about either of these quantities individually; for example, we’re not saying that it would be impossible to construct a perfectly adversarially robust classification model, only that, if you observe lots of errors in additive noise, you should not expect much robustness, and vice versa.
> >
> > You make a very good point about our use of the word “generalization”; we do indeed use this in several places where “test error” would be more appropriate. We will definitely fix this in future drafts. Moreover, regarding the tradeoff you mentioned, the test error we are most concerned with in this paper is test error under additive noise, not test error on the clean image distribution. We don’t intend to claim anything about a relationship between clean test error and adversarial robustness, but we can see a few places in the paper, especially the introduction, where a reasonable reader might think that we did. We will be sure to address this in our revisions.
> >
> > You asked about the definition of the error set. There are two parts to the answer. First, when considering images from the test set under additive noise, we always assume that the “correct” label of the noisy point is the same as that of the clean point; we ran all our experiments on scales of noise at which this is reasonable. We agree that the paper should be edited to make this clearer.
> >
> > The second part of the answer is that, for all figures and tables, a point is an “error” if the largest logit belongs to an incorrect class regardless of the confidence. However, the isoperimetric inequality is just a fact about sets in R^n, and could just as easily apply to the set of “confident errors” made by a model. In practice, we found that this makes very little difference to the overall story; the error sets we examined all contain less confident errors on the boundary and more confident errors very closeby in the interior. (There are in fact some visualizations of this in the appendix.) We agree, though, that this might be worth discussing in the main body of the paper.
> >
> > Thank you for pointing out the terminology overload with “typical set”; we can certainly be more careful about this.
> >
> > You say: “It is unnecessary to suggest that the only way to defend against small perturbations is to reduce the error rate in Gaussian noise, this is only one way that provably doesn’t scale.” We are not entirely sure what you are asking here, but if you are saying that Gaussian data augmentation isn’t the only way to improve robustness, then we agree. The Gaussian isoperimetric inequality merely implies that if you have not improved robustness to Gaussian noise then you have not improved adversarial robustness beyond the bound given by the theorem. This has nothing to do with any particular training procedure; in fact, what you say --- “approaches which improve adversarial robustness also improve robustness to Gaussian noise” --- is exactly what we meant.
> >
> > Regarding the question about failed defenses not improving generalization in Gaussian noise, we again mean the thing that you suggest. You are of course correct that adding noise can only make the problem harder; the thing to notice in Table 2 is that the failed defenses, unlike adversarial training, did not perform better than the *undefended* model on noisy images. The general takeaways from Tables 1 and 2 are (a) the defense that seems to improve adversarial robustness also showed improvement (over the undefended model) in most of the noise distributions we tested, (b) failed defenses don’t, and (c) training on noise seems to improve adversarial robustness.
> >
> > What do you mean by “It is difficult to draw definitive conclusions from Tables 1 and 2 because the random error was not reported.”? What is “random error”? We do report error rates on the clean test set in the rows labelled “Clean”.
> >
> > Thanks again for reading, and let us know if there’s anything else we can help clarify or you find any other mistakes.

---

> > > ### Public Comment · ~Angus_Galloway1 · 2018-10-06
> > > **Clarification, and re. the conclusion about previously proposed defenses**
> > >
> > > Thanks for the detailed response.
> > >
> > > I mention the literature re. clean accuracy vs robustness trade-off because it applies to absolute statements that appear in the context of reducing the error rate in Gaussian noise to zero, and “imperfect generalization”. I do not believe that making any error rate zero is practical, nor that obtaining this for Gaussian noise is desirable. Perhaps some of the absolute statements can be clarified.
> > >
> > > Regarding “The only way to defend against such adversarial perturbations is to reduce the error rate in Gaussian noise”, I understand what was meant now, and initially interpreted this as suggesting that the only/best way to do this is also by training on Gaussian noise, as this is one of the strategies explored in the paper. I agree that testing on additive Gaussian noise is a good idea, but the best way to reduce the error rate in this noise is not likely to be by training on the same, similarly to how adversarial training is generally not an optimal way to reduce epsilon-adversarial test error while improving generalization (although the reasons for the respective approaches being suboptimal are unique).
> > >
> > > The differences in test accuracy for the previously proposed defenses benchmarked in Table 2 are rather small, and the statement that none of them improve generalization in noise should be supported by several independent runs to show the variance of the results to help establish if the differences are significant. There are different ways of expressing the random error, one way would be to repeat the experiments over n=5 random seeds and report the mean to a number of decimal places given by the precision, where precision could be the standard error of the mean for n samples and is usually rounded to 1 sig fig. This repetition should still be done even if the differences do not seem small, and is especially important when drawing conclusions about several different methods. I could be convinced that this might be less important when sweeping a single variable where a clear trend can be observed.
> > >
> > > Setting aside the issue of random fluctuations, Table 2 shows that “tv-min” does 3.1% (absolute) better than the undefended InceptionV3 model for sigma=0.1, and 14.9% (absolute) better for sigma=0.2. If we’re comparing to the undefended InceptionV3 model, test performance on additive noise is approximately upper bounded by that for the clean test set (76%) by the data processing inequality, so an increase from 68.4% to 71.5% for sigma=0.1 represents about 40% of the available room for improvement for this model, and 62% in terms of the same for sigma=0.2. The approximate upper bound is exact if we assume that the undefended model was trained to maximize test accuracy and the defenses do not increase the effective capacity. Only one of the defenses exceeds this upper bound ("HGD" by 1.2%), so these seem like reasonable assumptions, but again difficult to know without random error. “Pixel” also achieves higher accuracy than the baseline for the noisy datasets, although the improvement is less than for “tv-min”. Is there a reason these defenses are being excluded? Otherwise I believe the statement that none of the defenses improve the test error in noise needs adjustment. Some of the defenses seem to reduce generalization error, e.g., “Random”, because it has less clean test accuracy for the same accuracy in additive noise sigma=0.2 as the baseline. I'm assuming this is one of the places where it was previously acknowledged that "test error" would be more appropriate than "generalization".
> > >
> > > Sorry to insist on so many details, i'm mainly seeking clarity as to the recommendations/results, and hoping to give the previously proposed defenses a fair shake even if they aren't optimal.

---

> > > > ### Author Response · Authors · 2018-10-08
> > > > **clarification about the tradeoff and recommendations**
> > > >
> > > > Hi, Angus,
> > > >
> > > > Thanks again for the comments. We’re still not exactly sure how the work about trading off clean accuracy for robustness is relevant to the main claims of our paper. At least for the noisy image distribution that we considered in Section 4, we prove a theorem that shows that the existence of test error *implies* the existence of small worst-case l2 perturbations. For example, if sigma=.1, an error rate of 10^-10 implies that the median distance to the nearest error is at most -.1 * phi^{-1}(10^{-10}) = .624. We agree that getting zero test error does not seem practical, but unfortunately, at least for the noisy image distribution, our theorem implies that obtaining the robustness results that the defense literature seems to be aiming for requires the test error to be very close to zero.
> > > >
> > > > None of the work on the tradeoff between accuracy and robustness applies directly to the case we consider here, and we are in fact arguing that in our case the opposite is true: our experiments showed that lower error rates in additive noise do seem to correlate with increased robustness. This doesn’t contradict the “tradeoff” literature you are citing; it’s just that we are discussing a different problem --- they are concerned with accuracy on the clean test set where we are concerned with accuracy in the presence of additive noise.
> > > >
> > > > In general, we don’t have a recommendation for how to solve either the adversarial example problem or the problem of increasing accuracy under additive noise. Rather, we are saying that these are provably the same problem for the noisy image distribution, and they seem to be the same problem for the clean image distributions we examined. In light of this, we believe that completely solving the adversarial example problem is probably going to be incredibly difficult.
> > > >
> > > > It has been helpful to talk to you about your reactions to the paper; it has shown us a few places where we think we can make the exposition clearer. Please let us know if there’s anything else we can answer.

---

> > > > ### Author Response · Authors · 2018-10-26
> > > > **Updating the defense table.**
> > > >
> > > > Hi Angus,
> > > >    Just an update, we reran each of the defenses N=10 times and took the mean accuracies for each and computed the standard error of these measurements. We found that only one of the defenses (Pixel Deflection) showed any statistically significant improvement over the baseline undefended model, the rest actually increase test error in additive noise. For Pixel Deflection the improvement was still minimal when compared with what can be achieved by performing standard Gaussian data augmentation. For sigma=.1 Pixel deflection had a mean accuracy of 70.3% with standard error of 1.1 while the undefended model (InceptionV3) mean accuracy was 69.1%.  Gaussian data augmentation achieved 73.3% accuracy.
> > > >
> > > > We also will correct a mistake we made when transcribing the numbers for tv-minimize in the original table. The mean accuracies for this defense at sigma=.1 is 68% and at sigma=.2 is 50.5%. Both values are lower than the undefended baseline. The original table incorrectly reported higher numbers for this particular defense. We will update the paper with the exact numbers, using a bar plot to visualize the mean accuracies and corresponding standard errors more easily.

---

> > > > > ### Public Comment · ~Angus_Galloway1 · 2018-11-02
> > > > > **Thank you**
> > > > >
> > > > > Thanks for responding to these concerns, I look forward to the updated version.
> > > > >
> > > > > I maintain that the trade-off between clean test accuracy and robustness still applies despite the additive white noise, as this cannot "cancel out" the noise inherent in the dataset wrt the label without significant destruction of the relevant information that describes the rule to be learned.

---

### Author Response · Authors · 2018-11-13
**Rebuttal**

Thank you to all three reviewers for taking the time to read and comment on our paper. We would like to clarify the significance of our results.

Many members of the machine learning community treat the existence of small worst-case perturbations as surprising, while far fewer would react in the same way to the fact that vision models perform imperfectly in the presence of noise. Some authors have tried to blame the existence of nearby errors on strange properties of the error set itself. We argue that there is neither any evidence for any such strangeness nor any need to invoke it to explain the results we see: given the error rates we observe in noise, an error set whose boundary is almost linear, combined with some unintuitive but mathematically straightforward facts about high-dimensional geometry, is already enough to predict the existence of adversarial examples at the scales we see in practice.

Our goal is not to propose yet another adversarial defense. Rather, it is to demonstrate that, given the (unsurprising) performance of vision models on noisy images, one should in fact not be surprised that there are also errors very close to a clean image. We show that the measurement we propose --- measuring the model’s performance on noisy perturbations of test images --- is (a) highly correlated with adversarial robustness, (b) easier to measure in practice, and (c) interesting in its own right.

We establish the relationship between these two quantities --- distance to the nearest error and error rate in noisy image distributions --- in two different settings. In the first setting, where we use a Gaussian centered at a test point as the image distribution, the relationship is tight: for a given error rate, the Gaussian isoperimetric inequality gives a model-independent bound on how far away the nearest error can be from a typical sample. (To address a question that came up a few times in the reviews, we assume throughout the entire paper that the noise scales we consider are small enough not to change the label; some pictures of noisy images at different scales can be found in the appendix. There have been numerous studies of the performance of machine learning models under various image corruptions which all make this assumption[1,2,3,4,5,6].) Moreover, we saw that, for the error rates in noise that we observe in practice, the distance from most noisy points to the nearest error is not far from the bound. So, in this image distribution, it is impossible to improve adversarial robustness without simultaneously reducing test error.

In the second setting, where we use the ordinary distribution, we can’t give a hard bound of this type, but we can establish the relationship between the two quantities of interest empirically. (In particular, to answer reviewer #1’s last question, we are not claiming to somehow use the Gaussian isoperimetric inequality from the previous section to deduce a bound for the clean point.) But the empirical case is quite strong: in every case we examined, models which showed improvement in one of these two metrics also showed improvement in the other. This was true even when, as in the case of adversarial training or training on noise, we intervened in an attempt to improve one and not the other. (To address a question from reviewer #3, the models trained in noise used the same number of samples as the ordinarily trained models, but again, the point of this experiment was to establish the relationship between the two quantities of interest, not to propose a new defense.) We also show, in Table 1, that this remains true when we examine a few other noise distributions, and, in Table 2, we looked at six broken adversarial defenses and saw that measuring performance in noise would have been enough to identify that they did not work.

Most adversarial example work operates under the assumption that the errors they are trying to eliminate are “special” in some way, and that the task of eliminating them is different from the task of reducing test error. We are arguing for a different perspective: these nearby errors are not surprising or complicated, and are in fact a natural consequence of test error in noise. The situation is, in a sense, much simpler than commonly assumed: the defense community and the community studying model robustness to additive noise are both trying to reduce test error; they are just measuring it in two different ways.

https://openreview.net/pdf?id=HJz6tiCqYm
https://arxiv.org/abs/1705.02498
https://arxiv.org/pdf/1604.04326.pdf
https://arxiv.org/pdf/1706.06969.pdf
https://arxiv.org/pdf/1611.05760.pdf
https://arxiv.org/pdf/1703.08119.pdf

---

### Meta-Review · Area_Chair1 · 2018-12-16
**Major revisions required.**

**Confidence:** 4
**Recommendation:** Reject

**Metareview:**

In light of the reviews and the rebuttal, it seems that the paper needs to be rewritten to head off some of the confusions and criticisms that the reviewers have made. That said, the main argument seems to contradict some of the lower bounds recently established by Madry and colleagues, showing the existence of distributions where the sample complexity for finding robust classifiers is arbitrarily larger than that for finding low-risk classifiers. I recommend the authors take a closer look at this apparent contradiction when revising.

---

> ### Public Comment · ~Dimitris_Tsipras1 · 2018-12-23
> **No contradiction with mentioned work**
>
> I am an author of the mentioned work (https://arxiv.org/abs/1804.11285). I would like to say that there is no contradiction between the results of our paper and this one. One can adapt our bounds separating standard learning from adversarially robust learning to separate standard learning from large-noise-robust learning.
>
> Learning a classifier that is robust to noise (of appropriately large magnitude) is harder in terms of sample complexity compared to learning a standard low-risk classifier.

---

> > ### Comment · Area_Chair1 · 2018-12-23
> > **thanks for clarification**
> >
> > Thanks for the clarification. Just to be clear, this "apparent contradiction" was just something I wanted the authors to engage with. It's not mentioned by any of the referees, who raise different issues. I am certain that a rewrite of this paper (designed to anticipate confusion that will arise by readers, such as those by these referees) will produce a much stronger and more impactful paper.
> >
> > Regarding the apparent contradiction, I was imagining a separation between robust-to-adversary and robust-to-large-noise learning, which you do not address. Can you comment on this?

---

> > > ### Public Comment · ~Dimitris_Tsipras1 · 2019-02-11
> > > **Clarification**
> > >
> > > Apologies for the late response, I did not receive a reply notification.
> > >
> > > No such separation between robust-to-adversary and robust-to-large-noise learning arises from the results of our paper. Note that the optimal classifier for the Gaussian setting is a linear classifier. For linear classifiers, adversarial robustness and robustness to (large) noise are both directly dependent on the margin achieved by the classifier and are hence very closely connected.